# Arithmetic-Mean $\mu$P for Modern Architectures: A Unified Learning-Rate Scale for CNNs and ResNets

## Abstract

Choosing an appropriate learning rate remains a key challenge in scaling depth of modern deep networks. The classical maximal update parameterization ($\mu$P) enforces a fixed per-layer update magnitude, which is well suited to homogeneous multilayer perceptrons (MLPs) but becomes ill-posed in heterogeneous architectures where residual accumulation and convolutions introduce imbalance across layers. We introduce *Arithmetic-Mean $\mu$P* (AM-$\mu$P), which constrains not each individual layer but the *network-wide average* one-step pre-activation second moment to a constant scale. Combined with a residual-aware He fan-in initialization—scaling residual-branch weights by the number of blocks ($\mathrm{Var}[W] = c/(K \cdot \mathrm{fan\text{-}in})$)—AM-$\mu$P yields width-robust depth laws that transfer consistently across depths. We prove that, for one- and two-dimensional convolutional networks, the maximal-update learning rate satisfies $\eta^\star(L) \propto L^{-3/2}$; with zero padding, boundary effects are constant-level as $N \gg k$. For standard residual networks with general conv+MLP blocks, we establish $\eta^\star(L) = \Theta(L^{-3/2})$, with $L$ the minimal depth. Empirical results across a range of depths confirm the $-3/2$ scaling law and enable zero-shot learning-rate transfer, providing a unified and practical LR principle for convolutional and deep residual networks without additional tuning overhead.

## 1 Introduction

Training deep networks is highly sensitive to the learning rate (LR). In homogeneous MLPs, "maximal update" ($\mu$P) principles yield width-robust LR settings that transfer across depth by keeping the one-step pre-activation variance at a constant scale. Modern architectures, however, are dominated by *residual networks (ResNets)* and *convolutional networks (CNNs)*, where residual accumulations render layer statistics inherently heterogeneous and convolutions introduce spatial–channel coupling and boundary effects (circular vs. zero padding). Enforcing identical per-layer update magnitudes (e.g., setting each layer's update variance to 1) is overly restrictive for such heterogeneous networks; a network-level budget is more appropriate.

**A more general $\mu$P LR: network-wide scale (AM-$\mu$P).** Denote $L$ by the *minimal effective depth* (each residual block counts as one depth unit; intra-block sublayers only induce lower-order corrections). For input $x$ and layer $\ell$, write the one-step pre-activation update as $\Delta z_i^{(\ell)}(x)$ and define the per-layer second moment

$$S_\ell \; := \; \mathbb{E}_{x \sim \mathcal{D}}\big[(\Delta z_i^{(\ell)}(x))^2\big].$$

Instead of forcing $S_\ell = 1$ for all $\ell$, we *fix the network-wide average* to a constant scale:

$$\bar{S} \; := \; \frac{1}{L}\sum_{\ell=1}^{L} S_\ell \; = \; 1,$$

which upgrades $\mu$P from a "per-layer equal-amplitude" rule to a *network-level budget* that remains width-robust while allowing layers to reallocate update magnitudes under residual accumulation, convolutions, and boundary effects. In homogeneous cases it reduces to the classical $\mu$P criterion.

**Residual-aware initialization.**   We pair the above LR scale with a residual-aware He initialization: for a model with $K$ residual blocks, we scale residual-branch weights with variance

$$\text{Var}[W] \;=\; c/\bigl(K \cdot \text{fan\_in}\bigr),$$

which keeps forward/backward second moments controlled across depth.

**Main results.**   Within this unified initialization and LR scale, we analyze **1D/2D CNNs** (handling both circular and zero padding) and **standard ResNets** (identity-type skips as the default; a few projection/downsampling shortcuts contribute only constant/boundary-level corrections). Residual blocks are allowed to contain general *conv+MLP* substructures (not merely a single MLP layer).

*CNNs and MLPs.* For 1D/2D CNNs,

$$\eta^\star(L) \;\propto\; L^{-3/2}.$$

With *circular padding*, visits are uniform and the recursion mirrors the fully connected case. With *zero padding*, boundary non-uniformity introduces corrections proportional to boundary ratios; when the spatial width $N$ is *much larger* than the kernel's effective coverage $k$ (i.e., $N \gg k$), these corrections become constant-level and do not change the leading $L^{-3/2}$ law.

*ResNets (general residual blocks).* For standard ResNets,

$$\eta^\star(L) \;=\; \Theta\bigl(L^{-3/2}\bigr).$$

Compared to the proportional form for CNN/MLP, residual accumulation and layer-wise heterogeneity make the constant characterization more conservative (hence $\Theta(\cdot)$), while preserving the same order in $L$.

**Implications.**   Under AM-$\mu$P and a residual-aware initialization, **CNNs align with MLPs** to the proportional $L^{-3/2}$ depth law, whereas **ResNets match the order** but with a more conservative constant characterization. For zero padding, the engineering condition $N \gg k$ gives a verifiable regime where boundary effects are constant-level.

**Empirical validation.**   On homogeneous CNN/ResNet families (ReLU/GELU, He fan-in, SGD without momentum, fixed batch size), we sweep LR on a logarithmic grid across depths $L$ and record the maximal-update LR $\eta^\star$. We observe: (i) a stable log–log slope near $-3/2$; (ii) zero-shot LR transfer across depths; (iii) activation changes affect constants but not the depth exponent; (iv) padding and width mainly affect constant factors. Full curves, ablations, and **additional results on CIFAR-100 and ImageNet** appear in the appendix.

**Contributions.**

- **A more general network-level $\mu$P LR scale.** We propose AM-$\mu$P, a network-level update-budget criterion that is equivalent to classical $\mu$P in homogeneous settings and remains valid under residuals/convolutions/boundaries.

- **Unified depth–LR laws.** With the above scale and initialization we prove $\eta^\star(L) \propto L^{-3/2}$ for CNNs/MLPs and $\eta^\star(L) = \Theta(L^{-3/2})$ for ResNets; we systematically treat circular vs. zero padding and the sufficiency of $N \gg k$.

- **General residual blocks.** Residual blocks may contain conv+MLP sublayers (not just a single MLP), yet the depth laws and cross-depth transfer persist.

- **Practice-oriented guidance.** Experiments across depths and activations corroborate the $-3/2$ slope and zero-shot transfer, providing direct LR-setting guidance for large-scale training.

**Organization.**   Subsection 3.1 and subsection 3.2 formalize the model and the AM-$\mu$P scale. Subsection 3.3 presents the CNN (1D/2D; circular/zero) results and boundary/finite-width corrections. Subsection 3.4 establishes the ResNet $\Theta(L^{-3/2})$ law under general residual blocks. Section 4 reports experiments and ablations. The appendix contains full proofs and additional experiments, including CIFAR-100 and ImageNet, among others.

## 2 RELATED WORK

### 2.1 NEURAL NETWORK INITIALIZATION AND UPDATE SCALE CHALLENGES

Stable training of deep neural networks critically depends on the interplay between weight initialization and the scale of parameter updates. Classical schemes such as Xavier initialization (Glorot & Bengio, 2010) and He initialization (He et al., 2015) aim to preserve the variance of activations and backpropagated gradients across layers, thereby mitigating vanishing or exploding signals. These methods are particularly effective for specific architectures—for example, He initialization in ReLU networks and its extensions to convolutional layers and residual structures (Taki, 2017)—but they primarily address stability at the initialization stage.

However, initialization alone cannot ensure consistent update magnitudes across layers during training, especially in modern architectures with residual connections, convolutions, or multiple pathways. Factors such as the number of signal paths, kernel sizes, and channel dimensions can cause substantial variation in update scales between layers, leading to imbalances between shallow and deep layers. Such imbalances may slow convergence or destabilize training, highlighting the need for a theoretical framework that explicitly controls update scales across the entire network. The next subsection introduces one representative approach—$\mu$P (Yang et al., 2022).

### 2.2 ORIGINAL $\mu$P REGIME FOR MLPS

$\mu$P was first proposed by Yang et al. in Tensor Programs V Yang et al. (2022) as a principled way to enable hyperparameter transfer across widths in MLPs. Its core idea is to select parameter initialization and global learning rate such that, for all hidden layers (except input and output), the per-layer pre-activation update variance remains $\mathcal{O}(1)$:

$$\mathbb{E}_{x \sim \mathcal{D}} \left[ (\Delta z_i^{(\ell)}(x))^2 \right] = 1, \quad \forall \ell.$$

This ensures that training dynamics are stable under width scaling, allowing hyperparameters tuned on small models to generalize to larger ones. The open-source `mup` library (Microsoft Research, 2022) provides a PyTorch interface for applying $\mu$P in practice.

Jelassi et al. (Jelassi et al., 2023) further investigated the depth dependence of $\mu$P learning rates in ReLU MLPs. Under mean-field initialization assumptions, they proved that while the critical learning rate $\eta^\star(L)$ is independent of width $n$, it scales with depth $L$ as

$$\eta^\star(L) \propto L^{-3/2},$$

revealing a nontrivial interaction between depth and stable update magnitudes. This result emphasizes the importance of depth-aware learning rate adjustment even under $\mu$P scaling.

Subsequent works have generalized the $\mu$P framework beyond plain MLPs. For instance, Chen et al. Chen (2024) proposed architecture-aware scaling methods compatible with residual and hybrid networks, and Chizat et al. Chizat et al. (2024) introduced the "Feature Speed Formula," offering a flexible theory for scaling hyperparameters in deep networks while recovering key $\mu$P properties.

In summary, the original $\mu$P regime provided a solid theoretical foundation for width scaling in MLPs. Later developments, particularly the discovery of depth dependence, laid the groundwork for adapting $\mu$P principles to more complex and realistic architectures.

### 2.3 INITIALIZATION FOR MLP, CNNS AND RESIDUAL NETWORKS

Weight initialization plays a critical role in enabling deep ReLU-activated networks to train effectively. Here we summarize three key approaches:

**He Initialization in MLPs** He et al. He et al. (2015) proposed initializing weights in fully-connected ReLU networks by sampling

$$W_{ij} \sim \mathcal{N}\left(0, \tfrac{2}{n_{\text{in}}}\right),$$

where $n_{\text{in}}$ is the input (fan-in) dimension. This simple strategy preserves the variance of activations and gradients across layers, significantly improving trainability in very deep networks.

**Scaled initialization for 1D/2D CNNs.** For a convolution with kernel support $\mathcal{K} \subset \mathbb{Z}^d$ ($d \in \{1, 2\}$) of size $|\mathcal{K}| = k$ and $C_{\text{in}}$ input channels, the effective fan-in is $n_{\text{in}} = k\, C_{\text{in}}$; adopting the rectifier-friendly He initialization (He et al., 2015) together with a mean-field "gating" factor $q := \mathbb{E}[\sigma'(z)^2]$ (see e.g., (Schoenholz et al., 2017; Xiao et al., 2018)) gives

$$W_{\text{conv}} \sim \mathcal{N}\Big(0,\ \frac{1}{q\, k\, C_{\text{in}}}\Big).$$

This single expression specializes to the usual 1D ($k$ is the kernel length) and 2D ($k = k_h k_w$) cases and preserves stable signal propagation in deep convolutional nets (cf. (Glorot & Bengio, 2010) for earlier schemes). For residual architectures, scaling with depth further improves stability (Taki, 2017; Zhang et al., 2019; De & Smith, 2020; Bachlechner et al., 2021).

**Scaled Initialization for ResNets** Taki (2017) analyzed simplified ResNet models and showed that their robustness to initialization hinges on appropriately scaling weight variance relative to the number of residual blocks. Specifically, initializing with

$$\text{Var}(W_{\text{res}}) = \frac{c}{K\, n},$$

where $K$ is the number of residual blocks, $n$ is the layer fan-in, and $c = O(1)$, helps preserve signal and gradient stability even in very deep residual architectures (Taki, 2017).

## 3 METHODS

### 3.1 PRELIMINARIES: $\mu$P REGIME EXTENSION

To enable hyperparameter transferability across model widths, the $\mu$P (maximal-update parameterization) regime fixes a global learning rate so that a layerwise pre-activation update has $\mathcal{O}(1)$ magnitude under width scaling. In its original MLP form, one enforces at a reference layer $\ell$:

$$\mathbb{E}_{x \sim \mathcal{D}}\Big[(\Delta z_i^{(\ell)}(x))^2\Big] = 1.$$

Modern architectures (skip/residual, convolutional branches) induce heterogeneous per-layer update scales, so single-layer control becomes inadequate. We therefore extend $\mu$P to a network-wide constraint.

**AM-$\mu$P Regime** Let $L$ denote the minimal effective depth. Define the layerwise update variance

$$S_\ell \equiv \mathbb{E}_{x \sim \mathcal{D}}\Big[(\Delta z_i^{(\ell)}(x))^2\Big], \qquad \bar{S} \equiv \frac{1}{L}\sum_{\ell=1}^{L} S_\ell.$$

We say the network is in the *AM-$\mu$P regime* if

$$\bar{S} = 1.$$

**Rationale for the arithmetic mean (formal rationale in Appx. A)** (i) *Reduction to original $\mu$P.* In homogeneous layers ($S_\ell \approx S$), $\bar{S} = 1$ implies $S_\ell \approx 1$ for all $\ell$. (ii) *Global scale control.* By AM bounds, $\min_\ell S_\ell \leq \bar{S} \leq \max_\ell S_\ell$, so the overall update scale is $\mathcal{O}(1)$ despite heterogeneity. (iii) *Network-wide consistency.* The constraint lifts the maximal-update principle from a single layer to the whole network, aligning with residual/skip compositionality.

Unless otherwise stated, all subsequent results and experiments are based on this extended definition; formal uniqueness/robustness justifications (A1–A7) and comparisons to geometric/harmonic means are deferred to Appendix A.

### 3.2 STRUCTURAL ASSUMPTIONS FOR CNNs AND RESIDUAL BLOCKS

We consider two families of architectures: (i) plain CNNs composed of homogeneous convolutional blocks (HCBs; detailed next), and (ii) pre-activation residual networks with identity skip connections (see *Residual Blocks* below). For any layer $\ell$, let $C_\ell$ denote the number of output channels, $\Lambda_\ell$ the

spatial index set, $N_\ell := |\Lambda_\ell|$ the spatial length, and $k_\ell := |\mathcal{K}_\ell|$ the kernel size. Convolutions use stride $s_\ell \equiv 1$; unless otherwise stated, we adopt circular padding so that feature maps are spatially stationary and $N_\ell = N_{\ell-1}$ within a block. We allow $\{C_\ell, k_\ell, N_\ell\}$ to vary with $\ell$.

To unify notation, we define the *effective width* of a convolutional layer as $M_\ell := C_\ell N_\ell$ (the total number of channel–position units). For fully-connected layers, the width is the number of neurons $n_\ell$. When we refer to "width-invariant" scaling, "width" means $M_\ell$ for CNNs and $n_\ell$ for fully connected layers. (Departures from exact homogeneity—e.g., zero padding or mild channel heteroscedasticity— will be treated as small corrections quantified later by $O(\max_\ell k_\ell / N_\ell) + O(\max_\ell 1/C_\ell)$.)

**CNNs.** Let spatial dimension $d \in \{1, 2\}$ with index set $\Lambda_\ell \subset \mathbb{Z}^d$ and size $N_\ell := |\Lambda_\ell|$ (in 2D, $N_\ell = H_\ell W_\ell$). Each convolutional layer $\ell$ has a kernel offset set $\mathcal{K}_\ell \subset \mathbb{Z}^d$ (arbitrary shape) with cardinality $k_\ell := |\mathcal{K}_\ell|$. With circular padding and stride 1, joint ranges of $p \in \Lambda_\ell$ and $\Delta \in \mathcal{K}_\ell$ visit each previous-layer site *exactly $k_\ell$ times* (torus indexing). Activation is ReLU, $\sigma(u) = \max(0, u)$, which satisfies $\sigma'(u)^2 = \sigma'(u)$. *Here $k_\ell$ denotes the kernel **cardinality**, whereas we will use $s_{\ell,r} := \max_{\Delta \in \mathcal{K}_\ell} |\Delta_r|$ for the axial half-span along axis $r$.*

Weights across different layers and indices are independent with zero mean. For a convolutional layer with $C_{\ell-1}$ input channels we use He fan-in initialization written via kernel cardinality:

$$\mathrm{Var}\big(W_{j,i,\Delta}^{(\ell)}\big) = \frac{2}{C_{\ell-1}\, k_\ell}, \qquad \Delta \in \mathcal{K}_\ell.$$

Equivalently, with $n_{\mathrm{in}} = C_{\ell-1} k_\ell$, the general form $1/(q\, n_{\mathrm{in}})$ (for a generic activation with $q = \mathbb{E}[\sigma'(z)^2]$) reduces to $2/n_{\mathrm{in}}$ for ReLU since $q = \frac{1}{2}$. Fully-connected layers (including those following the CNN) use

$$\mathrm{Var}\big(W_{j,i}^{(\ell)}\big) = \frac{2}{n_{\ell-1}},$$

and all biases are zero (or are independent with zero mean).

We assume a mild scale separation so that higher-order spatial covariance terms are negligible: along each spatial axis, the feature-map side length dominates the kernel extent. Using the axial spans $s_{\ell,r}$, we require $\min_r N_{\ell,r} \gg \max_r s_{\ell,r}$ (equivalently, $N_\ell^{1/d} \gg \mathrm{diam}(\mathcal{K}_\ell)$ in typical compact-kernel regimes), and channel widths are not pathologically small. In practice, $C_\ell \in [64, 512]$ with small kernels (e.g., 3–7 along each axis) and $H_\ell, W_\ell$ in the tens to hundreds usually satisfy this condition.[1]

*Remark (specializations and boundary effects).* (1) In 1D, $N_\ell$ is the sequence length and $k_\ell = |\mathcal{K}_\ell|$ is the kernel width; the above reduces to the standard $2/(C_{\ell-1} k_\ell)$ rule. (2) In 2D with a rectangular stencil, $k_\ell = k_{\ell,h} k_{\ell,w}$ and the variance becomes $2/(C_{\ell-1} k_{\ell,h} k_{\ell,w})$. (3) Replacing circular padding by zero padding breaks uniform coverage only near the boundary; the layerwise identities acquire $O\big(\sum_{r=1}^d s_{\ell,r}/N_{\ell-1,r}\big)$ corrections, which vanish as feature maps grow and do not affect leading-order scaling.

**Residual Blocks.** We consider pre-activation residual blocks with identity skip connections. Let $z_{\ell-1}$ denote the input to the $\ell$-th block and $z_\ell$ its output:

$$z_\ell = z_{\ell-1} + \mathcal{F}_\ell(z_{\ell-1}),$$

where the residual branch $\mathcal{F}_\ell$ is a composition of $m_\ell \geq 1$ layers of the form "linear map $\to$ ReLU", i.e.,

$$\mathcal{F}_\ell = T_\ell^{(m_\ell)} \circ \sigma \circ T_\ell^{(m_\ell-1)} \circ \cdots \circ \sigma \circ T_\ell^{(1)}, \quad \sigma(u) = \max(0, u).$$

Each linear map $T_\ell^{(t)}$ can be either a fully-connected layer or a 1D convolution (with constant kernel size, stride 1, and "same" padding to preserve the spatial length). Group or dilated convolutions are allowed, but the number of groups and dilation rate are $O(1)$.

Residual blocks are assumed to have matching input and output shapes so that the skip connection is an identity map, i.e., $n_\ell = n_{\ell-1}$ and $C_\ell = C_{\ell-1}$. Dimension changes (e.g., downsampling or

---

[1] As in common CNN backbones (VGG/ResNet-style), channels are in the hundreds while kernels are small; extremely narrow layers or unusually large kernels may violate this assumption.

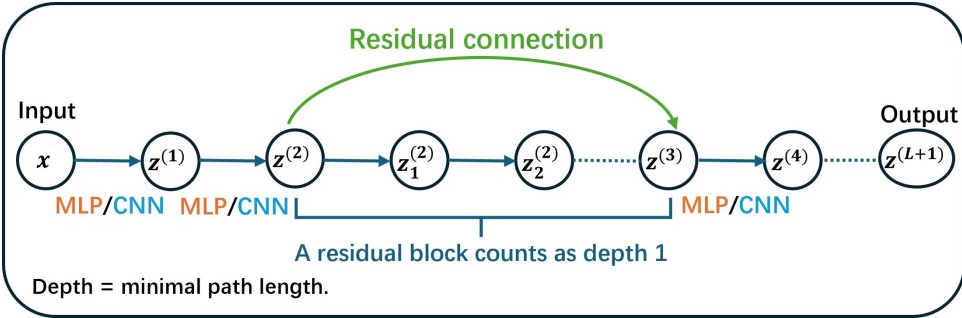

Figure 1: **Depth convention (minimal-path).** Depth equals the minimal path length; each residual block counts as 1.

channel projection) are allowed only in a vanishing fraction of blocks, whose total number is $O(1)$ relative to the total block count.

We define the *minimal depth $L$* of the network as the number of residual blocks, counting each block as one unit regardless of its internal depth $m_\ell$.see Fig. 1.

The internal depth is required to be sublinear in $L$, namely

$$\sup_\ell \frac{m_\ell}{L} \to 0,$$

so that per-block computations do not asymptotically dominate the scaling with respect to $L$.

These assumptions encompass standard ResNet architectures[2] : for example, the basic block corresponds to $m_\ell = 2$, while the bottleneck block corresponds to $m_\ell = 3$ (in a 1D analogue such as $1 \times 1$–$3 \times 1$–$1 \times 1$ convolutions). Dimension changes occur only in a small number of projection blocks, which can be accommodated within this framework.

### 3.3 Extension to Convolutional Networks

We extend the depth–learning-rate scaling to 1D/2D convolutional networks built from homogeneous convolutional blocks (HCBs; see Sec. 3.2). The results mirror the MLP case and show that convolution does not change the depth exponent, while also quantifying finite-width and boundary corrections that arise in realistic CNNs.

**Theorem 1** (Width-invariant depth scaling for homogeneous conv blocks in 1D/2D). *Let the spatial dimension be $d \in \{1, 2\}$ and consider a homogeneous convolutional block with stride $= 1$, circular padding, ReLU activation, and He fan-in initialization. For arbitrary channel widths $\{C_\ell\}$, arbitrary kernel supports $\mathcal{K}_\ell \subset \mathbb{Z}^d$ (of any size/shape), and spatial resolutions $\Lambda_\ell$ (so $|\Lambda_\ell| = N_\ell$ in 1D or $H_\ell W_\ell$ in 2D), the learning-rate scale that preserves width-invariant training dynamics satisfies*

$$\eta^\star(L) = \kappa\, L^{-3/2},$$

*where $\kappa$ depends only on the activation/initialization fixed point and is independent of $\{C_\ell, \mathcal{K}_\ell, \Lambda_\ell\}$.*

*Implication.* The exponent matches the MLP setting; thus convolutional structure does not alter the asymptotic depth dependence. A learning rate tuned at one width transfers to any other width without retuning. The proof is deferred to Appendix B.

**Theorem 2** (Finite-width, boundary, and mini-batch corrections in 1D/2D). *Under the setup of Theorem 1, but allowing mild departures from homogeneity (e.g., zero padding or mild channel heteroscedasticity) and mini-batch size $B$, the width-invariant depth scaling persists at leading order*

---

[2]This condition is consistent with standard ResNet designs: each residual block typically contains one or two ReLU–convolution operations. If $m_\ell$ is too large, the skip connection may lose its identity-like effect, reducing the inherent advantages of the residual structure. With the exception of special networks containing many skip connections (e.g., U-Net), most residual networks satisfy this sparsity assumption.

*and admits a uniform correction:*

$$\eta^\star\big(L; \{C_\ell, \Lambda_\ell, \mathcal{K}_\ell, B\}\big) = \kappa\, L^{-3/2}\left(1 + O\left(\underbrace{\max_\ell \frac{1}{C_{\ell-1}}}_{width} + \underbrace{\max_\ell \mathrm{bdry}(\Lambda_\ell, \mathcal{K}_\ell)}_{boundary} + \underbrace{\frac{1}{B}}_{batch}\right)\right),$$

*where the boundary fraction* $\mathrm{bdry}(\Lambda_\ell, \mathcal{K}_\ell)$ *quantifies the nonuniform coverage near the boundary induced by zero padding. Concretely:*

$$\mathrm{bdry}(\Lambda_\ell, \mathcal{K}_\ell) = \begin{cases} \dfrac{s_\ell}{N_\ell}, & \text{(1D), with } s_\ell := \max_{\Delta \in \mathcal{K}_\ell} |\Delta|, \\ \dfrac{s_{\ell,h}}{H_\ell} + \dfrac{s_{\ell,w}}{W_\ell}, & \text{(2D), with } s_{\ell,h} := \max_{\Delta \in \mathcal{K}_\ell} |\Delta_h|, \ s_{\ell,w} := \max_{\Delta \in \mathcal{K}_\ell} |\Delta_w|. \end{cases}$$

*In particular, these subleading terms do not alter the depth exponent* $-3/2$.

*Proof.* Deferred to Appendix B.

### 3.4 EXTENSION TO RESNET ARCHITECTURES

We now extend the depth–learning-rate scaling rule to ResNet architectures composed of standard residual blocks (see Sec. 3.2). Here, the network depth $L$ is measured as the *minimal depth*, meaning that each residual block counts as one depth unit regardless of the number of layers within it. Under the standing assumptions and adopting the AM-$\mu$P normalization across layers, we obtain the following result.

**Theorem 3** ($\mu$P scaling law for ResNets). *For a ResNet of minimal depth $L$ initialized with scaled He initialization* $\mathrm{Var}[w] = c/(Kn)$ *for $K$ residual blocks of width $n$, the learning-rate scale that preserves width-invariant training dynamics satisfies*

$$\eta^\star(L) = \Theta\big(L^{-3/2}\big).$$

This scaling law matches the exponent in the MLP setting, indicating that the residual connection structure does not alter the asymptotic depth dependence when depth is measured in minimal-depth units. Consequently, a learning rate tuned for a small-width ResNet can be transferred directly to any width without retuning. The proof is deferred to Appendix C.

## 4 EXPERIMENTS

We empirically validate the depth–learning-rate scaling laws for both CNNs and ResNets. For CNNs, we test Theorems 1 and 2; for ResNets, we test the $\mu$P-based counterpart stated in Theorem 3, which predicts the same asymptotic exponent $\eta^\star(L) \propto L^{-3/2}$ under scaled He initialization. We first describe the common protocol, then present results for homogeneous CNNs and on ResNets.

### 4.1 EXPERIMENTAL SETUP

**Datasets and metrics.** We use CIFAR-10 with standard train/val splits. For hyperparameter selection, we report top-1 *validation* accuracy; for final results, we report *test* accuracy.

**Protocol.** For each depth $L$, we sweep $\eta$ on a logarithmic grid and record the maximal-update learning rate $\eta^\star$ at the end of *one epoch*[3]. We then model the depth law on the log–log scale via

$$\log_{10} \eta^\star = \beta_0 - \alpha \log_{10} L + \varepsilon,$$

and report the fitted slope $\hat{\alpha}$ and $R^2$. When multiple measurements per depth are available (e.g., across random seeds), we show the depth-wise mean $\pm$ 95% confidence interval for $\log_{10} \eta^\star$ and fit the line using *weighted least squares* with weights inversely proportional to the estimated variance of the depth-wise mean; we also plot the 95% confidence band of the fitted line. Otherwise, we use ordinary least squares.

---

[3]We adopt a single-epoch proxy for efficiency and comparability, consistent with the architecture-aware scaling protocol (base maximal LR determined at one epoch) (Chen, 2024) and with the $\mu$P view that optimal LRs are governed by early-training dynamics (Jelassi et al., 2023).

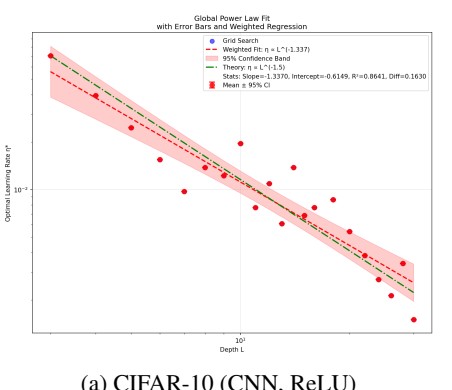

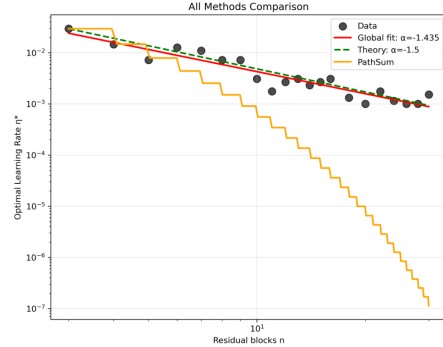

(a) CIFAR-10 (CNN, ReLU)

(b) CIFAR-10 (ResNet): Global fit, AM-$\mu$P theory, and PathSum (Chen (Chen, 2024))

Figure 2: **Global depth–LR scaling on CIFAR-10.** (a) CNN: grid-searched optima with 95% CIs and a weighted global fit. (b) ResNet: global fit alongside AM-$\mu$P theory and PathSum (Chen, 2024).

**Loss.** All CNN and ResNet experiments are trained with standard multi-class cross-entropy (mean reduction).[4]

### 4.2 CONVOLUTIONAL NETWORKS: UNIFIED EXPERIMENTS

**CNN-specific settings.**

- **Blocks.** Homogeneous 2D convolutional blocks; stride 1; circular padding unless stated.

- **Initialization & optimizer.** He fan-in initialization; SGD without momentum; batch size 128.

- **Depth counting.** $L$ counts `conv + nonlinearity` blocks; classifier: global pooling $\rightarrow$ linear head.

- **Ablations.** When specified, vary channel widths $\{C_\ell\}$, kernel sizes $\{k_\ell\}$, spatial resolutions $\{N_\ell\}$, and mini-batch size $B$ to probe finite-width/boundary/batch corrections.

- **Error Bars and Weighted Fit.** Mean $\pm 95\%$ CIs on $\log_{10} \eta^\star$ and a weighted least-squares global fit with its 95% confidence band (as specified in the Protocol).

**Learning-rate search and segmented prediction.** We sweep $\eta$ from $10^{-4}$ to $10^1$ (40 log-spaced points) and take $\eta^\star$ that maximizes validation accuracy after the proxy training. To test zero-shot depth transfer, we use a segmented baseline:

- Segment A: fit on $L \in \{3, 4\}$, predict $L \in \{5, \ldots, 9\}$.

- Segment B: fit on $L \in \{10, 11\}$, predict $L \in \{12, \ldots, 16\}$.

- Segment C: fit on $L \in \{18, 20\}$, predict $L \in \{22, \ldots, 30\}$.

We report $\hat{\alpha}$ (slope), intercept, and $R^2$, together with mean $\pm 95\%$ CIs for $\log_{10} \eta^\star$ and the 95% confidence band of the weighted global fit.

Across CIFAR-10 CNNs, the maximal-update learning rate $\eta^\star$ follows a clear power law in depth; the global fit yields a slope of about $-1.337$ (Fig. 2 **(a)**). We observe slightly larger dispersion at greater depths, which is consistent with finite-width, padding/boundary, and batch-variance effects primarily modulating the prefactor. Additional results (including CIFAR-100, ImageNet, GELU variants, segmented prediction and other architectural variants such as zero/circular padding) are provided in Appendix D.

---

[4]See Appx. F for why using CE in experiments is compatible with the MSE-based derivation.

### 4.3 ResNets: Scaling and Zero-shot Depth Transfer

**ResNet-specific settings.**

- **Architecture and depth.** We measure depth by the *minimal depth* $L$: each residual block counts as one unit, regardless of the number of layers inside. In plots we also report the *effective* depth $L_{\text{eff}} = 3L$ to align with CNN counting. Each unit contains two 3×3 conv layers (64 channels, stride 1, same spatial size) with an identity skip.

- **Initialization and optimizer.** Scaled He fan-in initialization; conv weights on the residual branches are multiplied by $1/\sqrt{K}$ for $K$ residual blocks to stabilize depth-wise variance (affecting the prefactor $\kappa$ but not the exponent). Optimization uses SGD without momentum; batch size 128.

- **Padding.** Circular padding unless otherwise noted; zero-padding comparisons are deferred to the appendix.

**Learning-rate sweep and segmented prediction (ResNet).** We use the same logarithmic LR grid as in the CNN section (40 points from $10^{-4}$ to $10^{1}$) and identify $\eta^\star$ after one epoch. For zero-shot transfer we fit a two-anchor line within each depth segment and predict $\eta^\star$ for held-out depths in that segment (anchor sets as displayed in the legend of Fig. 2(**b**)).

Across the evaluated depths, the maximal-update learning rate follows a clear power law: a global log–log fit yields $\hat{\alpha} = -1.435$, which closely matches our AM-$\mu$P prediction ($-1.5$) and indicates that residual connections do not alter the depth exponent (Theorem 3). In contrast, the PathSum curve (Chen, 2024) shows an increasing deviation from the empirical optima at larger depths. Two-anchor segmented fits transfer reliably within segments, whereas errors rise at segment boundaries and for the deepest models, consistent with finite-width and padding effects modulating the prefactor $\kappa$ rather than the exponent. Further results (e.g., CIFAR-100, ImageNet, and architectural variants including batch normalization and dropout) are provided in Appendix D.

## 5 Conclusion

We provide formal proofs that place CNNs and pre-activation ResNets on the same scaling footing. Under a *minimal depth* notion of depth (each residual block counts as one) and the CNN *effective width* $M_\ell = C_\ell N_\ell$, we *prove* a depth–learning-rate scaling law with exponent $-3/2$. The result is *tight for plain CNNs* under our assumptions (with explicit constants), and for ResNets we establish *order-level equivalence* via a minimal-depth reduction and block merge/split consistency; boundary and mild heterogeneity effects are quantified and shown to be lower order. *Crucially,* the law is width-invariant under our homogeneous-block view (arbitrary channel counts and kernel supports captured via $M_\ell$), providing a single depth currency across CNNs and pre-activation ResNets. These guarantees yield the following plug-and-play rule:

$$\eta^\star(L) = \eta^\star(L_0)\left(\tfrac{L}{L_0}\right)^{-3/2},$$

with the rest of the schedule unchanged, enabling one-time calibration at $L_0$ and drop-in transfer to arbitrary depths.

In standard SGD-family setups (ReLU/GELU activations, with or without BatchNorm/Dropout), the recipe scales cleanly to larger datasets—including CIFAR-100 and ImageNet—yielding robust cross-depth behavior and lower tuning cost in practice. By removing per-depth LR sweeps, the recipe streamlines experimental workflows, improving reproducibility and planning of compute budgets at scale.

**Outlook.** We will extend the unified scaling to Transformer/self-attention by formalizing an attention-block depth convention and aligning an "effective depth" with receptive-field/sequence-length growth, and by validating joint depth–LR (and sequence-length/field-of-view) scaling on long-sequence and multimodal tasks. Beyond CNNs/ResNets, AM-$\mu$P serves as a principled default for initial learning-rate selection in large-scale pretraining, providing a strong starting point to explore more efficient training protocols (e.g., reduced warmup, lighter sweeps, simplified schedules).

## REPRODUCIBILITY STATEMENT

We release complete source code and configuration files, along with detailed instructions for dataset acquisition, model training, and the comparison between grid-searched and theoretically derived learning rates. All theoretical proofs are presented in the Appendix with comprehensive explanations and explicit assumptions. We have thoroughly validated the implementation and have empirically corroborated the proposed AM-$\mu$P theory.

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

## A  RATIONALE FOR THE ARITHMETIC MEAN IN $\mu$P

*This appendix provides the formal rationale supplementing Sec. 3.1. We retain the notation $S_\ell$ and $\bar{S}$, and justify the arithmetic mean via (A1)–(A7), including failures of geometric/harmonic means and block-level split/merge consistency.*

**Scope and roadmap.** We formalize the choice of the arithmetic mean as a network-level aggregator by isolating structural axioms (permutation invariance, positive homogeneity, merge consistency), perturbation control under approximate orthogonality, and robustness under heterogeneity. We then document failure modes for geometric and harmonic means, establish block-level split/merge invariance at the effective-depth granularity, and verify consistency with the classical $\mu$P condition in the homogeneous limit.

Let $\Delta z^{(\ell)}(x)$ be the one-step pre-activation increment contributed by layer $\ell$ on input $x$. Define the layerwise energy

$$S_\ell := \mathbb{E}\Big[(\Delta z^{(\ell)}(x))^2\Big], \qquad \bar{S} := \frac{1}{L}\sum_{\ell=1}^{L} S_\ell.$$

The AM-$\mu$P design constraint fixes the *network-level* budget

$$\bar{S} \;=\; C \;=\; O(1).$$

**(A1) Additivity and merge-consistency (characterization).** Consider any partition $\{1,\ldots,L\} = \bigsqcup_{j=1}^{k} G_j$ with group means $m_j \triangleq |G_j|^{-1}\sum_{\ell\in G_j} S_\ell$. A network-level aggregator $\mathcal{M}$ should be (i) permutation-invariant, (ii) positively homogeneous $\mathcal{M}(cS) = c\,\mathcal{M}(S)$, and (iii) *merge-consistent*:

$$\mathcal{M}(S_1,\ldots,S_L) = \mathcal{M}(\underbrace{m_1,\ldots,m_1}_{|G_1|},\ldots,\underbrace{m_k,\ldots,m_k}_{|G_k|}) = \frac{\sum_{j=1}^{k} |G_j|\,m_j}{\sum_{j=1}^{k} |G_j|}.$$

Among symmetric means, these properties uniquely characterize the *arithmetic mean*. Hence, to respect additive layer energies and compositionality of subnetworks, we must take $\mathcal{M} = \bar{S}$.

**(A2) Truthful control of the total perturbation.** The total energy $\sum_{\ell=1}^{L} S_\ell = L\,\bar{S}$. When layer-wise increments are approximately orthogonal (as under residual/width normalizations),

$$\mathbb{E}\left[\left(\sum_{\ell=1}^{L} \Delta z^{(\ell)}\right)^2\right] = \sum_{\ell=1}^{L} S_\ell \; + \; 2\sum_{\ell < m} \mathrm{Cov}\left(\Delta z^{(\ell)}, \Delta z^{(m)}\right) \; \lesssim \; L\,\bar{S},$$

so fixing $\bar{S} = C$ pins the functional perturbation at $O(L)$ in second moment, yielding depth laws in one line thereafter.

**(A3) Robustness to heterogeneity (bounds and stability).** If $a \leq S_\ell \leq b$ (constant-factor heterogeneity), then $a \leq \bar{S} \leq b$. More generally, for any nonnegative weights $\{w_\ell\}$ with $\sum_\ell w_\ell = L$ (layer resampling),

$$\bar{S} = \frac{1}{L}\sum_\ell S_\ell = \frac{1}{L}\sum_\ell w_\ell\left(\frac{S_\ell}{w_\ell}\right) \; \geq \; \frac{L^2}{\sum_\ell \frac{w_\ell}{S_\ell}} \qquad \text{(Cauchy–Schwarz / Titu's lemma)},$$

showing AM control is not destabilized by a few extremely small/large layers (see (A4)/(A5)).

**(A4) Failure of geometric mean (GM): multiplicative cancellation.** Let $G = \left(\prod_\ell S_\ell\right)^{1/L}$. Take $S = (\varepsilon, \varepsilon^{-1}, \underbrace{1, \ldots, 1}_{L-2})$ with $\varepsilon \downarrow 0$. Then $G = 1$ remains constant while

$$\bar{S} = \tfrac{1}{L}\left(\varepsilon + \varepsilon^{-1} + L - 2\right) \to \infty,$$

so the total energy explodes and the global perturbation is not controlled. Moreover $\bar{S} \geq G$ (AM–GM), with equality only when all $S_\ell$ are equal; GM systematically underestimates in heterogeneous settings.

**(A5) Failure of harmonic mean (HM): hypersensitivity to small layers.** Let $H = \left(\frac{1}{L}\sum_\ell S_\ell^{-1}\right)^{-1}$. Then

$$\frac{\partial H}{\partial S_i} = \frac{H^2}{L} \cdot \frac{1}{S_i^2} > 0, \qquad S_i \downarrow 0 \; \Rightarrow \; \frac{\partial H}{\partial S_i} \uparrow \infty.$$

Maintaining a fixed $H$ forces disproportionate emphasis on small layers, distorting sensible layer-wise allocation. Also $H \leq \bar{S}$ (HM–AM), again biasing the total budget downward.

**(A6) Split/merge invariance at block level ("effective depth").** For a residual block $B$, define block energy $S_B = \sum_{\ell \in B} S_\ell$ and effective depth $K$ as the number of blocks. Any intra-block refinement (splitting a layer into sublayers) that preserves $S_B$ leaves the block-level AM $\frac{1}{K}\sum_{B=1}^{K} S_B$ unchanged. GM/HM, in contrast, generally change under the same split/merge, violating compositional consistency (cf. (A1)).

**(A7) Consistency with classical $\mu$P (degenerate homogeneous limit).** If $S_\ell \overset{d}{\approx} S$ (layerwise homogeneity), then $\bar{S} = C$ is equivalent to $S_\ell \approx C$ for all $\ell$. Thus AM-$\mu$P reduces to the original per-layer constraint in the homogeneous limit, while retaining linear control of $\sum_\ell S_\ell$ in heterogeneous architectures.

*Implication.* With $\bar{S} = C$, the subsequent derivations (given specific initialization/normalization) yield unified depth laws (e.g., $\eta^\star(L) \propto L^{-3/2}$) and block-level scaling that transfer across depths, while remaining compatible with residual-aware initializations.

# B    PROOF OF SCALING LAW FOR HOMOGENEOUS CONVOLUTIONAL BLOCKS

**Lemma 1** (Layerwise conditional expectation invariance (1D CNN, stride $= 1$)). *Under the structural assumptions in Sec. 3.2 (ReLU, stride $= 1$, circular padding, independent zero-mean weights*

*with fan-in variance), for any layer $h \in \{1, \ldots, L\}$ and any two parameter directions $\mu_1, \mu_2$, define*

$$T_h(\mu_1, \mu_2) := \frac{1}{C_h N_h} \sum_{j=1}^{C_h} \sum_{p \in \Lambda_h} \partial_{\mu_1} z_{j,p}^{(h)} \, \partial_{\mu_2} z_{j,p}^{(h)}.$$

*Then the following one-step invariance holds:*

$$\mathbb{E}\Big[ T_h(\mu_1, \mu_2) \, \Big| \, z^{(h-1)} \Big] = T_{h-1}(\mu_1, \mu_2).$$

*Consequently, by iteration,*

$$\mathbb{E}\Big[ T_L(\mu_1, \mu_2) \, \Big| \, z^{(\ell)} \Big] = T_\ell(\mu_1, \mu_2) \qquad \text{for all } \ell \leq L.$$

*Proof.* Fix $z^{(h-1)}$ and take expectation only over the weights of layer $h$. Let

$$a_{i,u}^{(1)} := \sigma'\big(z_{i,u}^{(h-1)}\big) \, \partial_{\mu_1} z_{i,u}^{(h-1)}, \qquad a_{i,u}^{(2)} := \sigma'\big(z_{i,u}^{(h-1)}\big) \, \partial_{\mu_2} z_{i,u}^{(h-1)}.$$

With stride $= 1$, the (pre-activation) derivative at layer $h$ expands as

$$\partial_\mu z_{j,p}^{(h)} = \sum_{i=1}^{C_{h-1}} \sum_{\Delta \in \mathcal{K}_h} W_{j,i,\Delta}^{(h)} \, a_{i,\,p+\Delta}^{(\mu)}.$$

Hence

$$\partial_{\mu_1} z_{j,p}^{(h)} \, \partial_{\mu_2} z_{j,p}^{(h)} = \sum_{(i_1, \Delta_1)} \sum_{(i_2, \Delta_2)} W_{j,i_1,\Delta_1}^{(h)} W_{j,i_2,\Delta_2}^{(h)} \, a_{i_1,\,p+\Delta_1}^{(1)} \, a_{i_2,\,p+\Delta_2}^{(2)}.$$

By independence and zero mean of distinct kernel parameters, only diagonal pairs survive under the conditional expectation:

$$\mathbb{E}\Big[ \partial_{\mu_1} z_{j,p}^{(h)} \, \partial_{\mu_2} z_{j,p}^{(h)} \, \Big| \, z^{(h-1)} \Big] = \sum_{i,\Delta} \mathrm{Var}\big(W_{j,i,\Delta}^{(h)}\big) \, a_{i,\,p+\Delta}^{(1)} \, a_{i,\,p+\Delta}^{(2)}.$$

Using the fan-in variance $\mathrm{Var}(W_{j,i,\Delta}^{(h)}) = \dfrac{2}{C_{h-1} k_h}$ with $k_h := |\mathcal{K}_h|$ and averaging over channels and positions,

$$\mathbb{E}\Big[ T_h(\mu_1, \mu_2) \, \Big| \, z^{(h-1)} \Big] = \frac{1}{C_h N_h} \sum_{j,p} \frac{2}{C_{h-1} k_h} \sum_{i,\Delta} a_{i,\,p+\Delta}^{(1)} \, a_{i,\,p+\Delta}^{(2)}.$$

The right-hand side is independent of $j$, so the factor $1/C_h$ cancels with $\sum_j$. By circular padding with stride $= 1$, when $p$ ranges over $\Lambda_h$ and $\Delta$ over $\mathcal{K}_h$, each $u \in \Lambda_{h-1}$ is visited exactly $k_h$ times. Thus

$$\frac{1}{N_h} \sum_p \sum_{\Delta \in \mathcal{K}_h} a_{i,\,p+\Delta}^{(1)} \, a_{i,\,p+\Delta}^{(2)} = \frac{k_h}{N_h} \sum_{u \in \Lambda_{h-1}} a_{i,u}^{(1)} \, a_{i,u}^{(2)}.$$

Since stride $= 1$ implies $N_h = N_{h-1}$, we get

$$\mathbb{E}\Big[ T_h(\mu_1, \mu_2) \, \Big| \, z^{(h-1)} \Big] = \frac{2}{C_{h-1} k_h} \cdot \frac{1}{N_h} \sum_{j=1}^{C_h} \sum_{p \in \Lambda_h} \sum_{i=1}^{C_{h-1}} \sum_{\Delta \in \mathcal{K}_h} \mathbf{1}\big\{ z_{i,\,p+\Delta}^{(h-1)} > 0 \big\} \, \partial_{\mu_1} z_{i,\,p+\Delta}^{(h-1)} \, \partial_{\mu_2} z_{i,\,p+\Delta}^{(h-1)}$$

$$= \frac{2}{C_{h-1} N_{h-1}} \sum_{i=1}^{C_{h-1}} \sum_{u \in \Lambda_{h-1}} \mathbf{1}\big\{ z_{i,\,u}^{(h-1)} > 0 \big\} \, \partial_{\mu_1} z_{i,\,u}^{(h-1)} \, \partial_{\mu_2} z_{i,\,u}^{(h-1)}.$$

$\square$

**Corollary** (Layerwise invariance in expectation). *Under the same structural assumptions and He+ReLU initialization, and either in the infinite-width limit or under the standard independence approximation between the ReLU gate and parameter-direction derivatives, we have the layerwise invariance*

$$\mathbb{E}\, T_h(\mu_1, \mu_2) = \mathbb{E}\, T_{h-1}(\mu_1, \mu_2), \qquad h = 1, \ldots, L.$$

**Remark.** *(i) The kernel size $k_h$ cancels exactly between the coverage count ($k_h$ visits per input position under circular padding, stride $= 1$) and the fan-in variance factor $1/k_h$, hence no explicit dependence on $k_h$ appears in the identity. Different kernel sizes across layers are therefore allowed. (ii) With non-circular padding, boundary positions are not visited uniformly; one obtains $\mathbb{E}[T_h \,|\, z^{(h-1)}] = T_{h-1} + O(s_h/N_{h-1})$, which vanishes for large feature maps, where $s_h := \max_{\Delta \in \mathcal{K}_h} |\Delta|$.*

**Lemma 2** (Second-moment decomposition of pre-activation changes in CNNs (top-layer form))**.** *Under the structural assumptions in Sec. 3.2 (ReLU, stride $= 1$, circular padding, independent zero-mean weights with fan-in variance), and assuming labels are independent of the network with $\mathbb{E}[y_{j,p;\alpha}] = 0$ and $\mathrm{Var}(y_{j,p;\alpha}) = \sigma_y^2$, for any depth $\ell \leq L$ and any channel–position pair $(i, p)$, after one SGD step*

$$\mathbb{E}\Big[(\Delta z_{i,p;\alpha}^{(\ell)})^2\Big] = A_{\mathrm{cnn}}^{(\ell)} \;+\; B_{\mathrm{cnn}}^{(\ell)}.$$

*Here*

$$B_{\mathrm{cnn}}^{(\ell)} = \sigma_y^2 \, \mathbb{E}\left[ \sum_{\mu_1, \mu_2 \leq \ell} \eta_{\mu_1} \eta_{\mu_2} \, \partial_{\mu_1} z_{i,p;\alpha}^{(\ell)} \, \partial_{\mu_2} z_{i,p;\alpha}^{(\ell)} \cdot \underbrace{\frac{1}{C_{L+1} N_{L+1}} \sum_{(j,p')} \partial_{\mu_1} z_{j,p';\alpha}^{(L+1)} \, \partial_{\mu_2} z_{j,p';\alpha}^{(L+1)}}_{=: \, T_{L+1}(\mu_1, \mu_2)} \right],$$

*and*

$$A_{\mathrm{cnn}}^{(\ell)} := \mathbb{E}\Bigg[ \sum_{\mu_1, \mu_2 \leq \ell} \eta_{\mu_1} \eta_{\mu_2} \, \partial_{\mu_1} z_{i,p;\alpha}^{(\ell)} \, \partial_{\mu_2} z_{i,p;\alpha}^{(\ell)}$$
$$\times \underbrace{\frac{1}{(C_{L+1} N_{L+1})^2} \sum_{(j_1, p_1)} \sum_{(j_2, p_2)} \big(\partial_{\mu_1} z_{j_1, p_1; \alpha}^{(L+1)} \, z_{j_1, p_1; \alpha}^{(L+1)}\big) \big(\partial_{\mu_2} z_{j_2, p_2; \alpha}^{(L+1)} \, z_{j_2, p_2; \alpha}^{(L+1)}\big)}_{=: \, S_{L+1}(\mu_1, \mu_2)} \Bigg].$$

**Proof.** By the chain rule,

$$\Delta z_{i,p;\alpha}^{(\ell)} = \sum_{\mu \leq \ell} \partial_\mu z_{i,p;\alpha}^{(\ell)} \, \Delta \mu, \quad \Delta \mu = -\eta_\mu \sum_{(j,p')} \partial_\mu z_{j,p';\alpha}^{(L+1)} \big(z_{j,p';\alpha}^{(L+1)} - y_{j,p';\alpha}\big).$$

Expand $(\Delta z_{i,p;\alpha}^{(\ell)})^2$, and take expectation over labels using $\mathbb{E}[y] = 0$, $\mathbb{E}[y^2] = \sigma_y^2$, independence across $(j, p)$ and from the network:

$$\mathbb{E}\big[(z_t^{(L+1)} - y_t)(z_s^{(L+1)} - y_s)\big] = z_t^{(L+1)} z_s^{(L+1)} \;+\; \sigma_y^2 \, \mathbf{1}\{t = s\}.$$

Collect the diagonal part ($t = s$) to obtain $B_{\mathrm{cnn}}^{(\ell)}$ with $T_{L+1}$; collect the off-diagonal and diagonal $z^{(L+1)} z^{(L+1)}$ part to obtain $A_{\mathrm{cnn}}^{(\ell)}$ with $S_{L+1}$. This yields the stated decomposition. $\qquad\square$

**Corollary** (Top-layer reduction via layerwise invariance)**.** *Under the assumptions of Lemma 2 and the Layerwise conditional expectation invariance (stride $= 1$),*

$$\mathbb{E}\Big[T_{L+1}(\mu_1, \mu_2) \,\Big|\, z^{(L)}\Big] = T_L(\mu_1, \mu_2), \qquad \mathbb{E}[T_{L+1}(\mu_1, \mu_2)] = \mathbb{E}[T_L(\mu_1, \mu_2)].$$

**Remark.** *When $C_\ell \equiv 1$ (single–channel), the channel–position averages in $T_{L+1}$ and $S_{L+1}$ reduce to width averages, and the lemma recovers the fully-connected formulas (cf. (Jelassi et al., 2023)); the residual case follows identically for homogeneous residual blocks with identity (or fixed scalar) skip connections.*

**Magnitude of the $A$-term.** By the definition of $S_{L+1}$ and weak dependence across channel–position indices, the dominant contribution in the double sum inside $S_{L+1}$ comes from $O(C_{L+1} N_{L+1})$ diagonal pairs, while off-diagonal terms do not change the order. Hence

$$\mathbb{E}\big[S_{L+1}(\mu_1, \mu_2)\big] = O\big((C_{L+1} N_{L+1})^{-1}\big), \qquad \mathbb{E}\big[T_{L+1}(\mu_1, \mu_2)\big] = O(1),$$

which implies

$$A_{\text{cnn}}^{(\ell)} = O\big((C_{L+1}N_{L+1})^{-1}\big).$$

Therefore we neglect $A_{\text{cnn}}^{(\ell)} = O((C_{L+1}N_{L+1})^{-1})$ in the width–spatial limit and focus on a recursive characterization of $B_{\text{cnn}}^{(\ell)}$.

From Lemma 2 and Lemma 1, we obtain

$$B_{\text{cnn}}^{(\ell)} = \sigma_y^2 \, \mathbb{E}\left[ \sum_{\mu_1,\mu_2 \leq \ell} \eta_{\mu_1}\eta_{\mu_2} \, \partial_{\mu_1} z_{i,p}^{(\ell)} \, \partial_{\mu_2} z_{i,p}^{(\ell)} \, T_\ell(\mu_1,\mu_2) \right].$$

Define the single–unit quantity

$$U_a^{(\ell)}(\mu_1,\mu_2) := \partial_{\mu_1} z_a^{(\ell)} \, \partial_{\mu_2} z_a^{(\ell)}, \qquad T_\ell(\mu_1,\mu_2) = \frac{1}{M_\ell} \sum_b U_b^{(\ell)}(\mu_1,\mu_2), \quad M_\ell := C_\ell N_\ell,$$

so that

$$B_{\text{cnn}}^{(\ell)} = \frac{\sigma_y^2}{M_\ell} \, \mathbb{E}\left[ \sum_{\mu_1,\mu_2 \leq \ell} \eta_{\mu_1}\eta_{\mu_2} \sum_{a,b} U_a^{(\ell)}(\mu_1,\mu_2) \, U_b^{(\ell)}(\mu_1,\mu_2) \right].$$

**Unit-wise equality.** In homogeneous CNNs (stride $= 1$, circular padding, channel i.i.d. and spatial stationarity), the relation

$$\mathbb{E}\big[U_a^{(\ell)}(\mu_1,\mu_2) \, T_{L+1}(\mu_1,\mu_2)\big] = \mathbb{E}\big[T_\ell(\mu_1,\mu_2)^2\big]$$

holds for every unit $a = (i,p)$. In practice, only the averaged form $\mathbb{E}[T_\ell T_{L+1}] = \mathbb{E}[T_\ell^2]$ is needed for the sequel. For non-circular padding or heterogeneous channels, the relation holds asymptotically with error terms $O(s_\ell/N_\ell) + O(1/C_{\ell-1})$, which vanish as feature maps grow.

**Overlap counting.** From the top-layer decomposition (Lemma 2) together with layerwise invariance (Lemma 1), we obtain

$$B_{\text{cnn}}^{(\ell)} = \sigma_y^2 \, \mathbb{E}\left[ \sum_{\mu_1,\mu_2 \leq \ell} \eta_{\mu_1}\eta_{\mu_2} \, T_\ell(\mu_1,\mu_2)^2 \right].$$

Grouping parameters by layer yields

$$B_{\text{cnn}}^{(\ell)} = \sigma_y^2 \, \mathbb{E}\left[ \sum_{h_1=1}^{\ell} \sum_{h_2=1}^{\ell} \sum_{\mu_1 \in \text{layer } h_1} \sum_{\mu_2 \in \text{layer } h_2} \eta_{\mu_1}\eta_{\mu_2} \, T_\ell(\mu_1,\mu_2)^2 \right],$$

and repeated invariance implies

$$\mathbb{E}\big[T_\ell(\mu_1,\mu_2)^2\big] = c_{\text{cnn}} \cdot \min\{h_1,h_2\}, \qquad (\mu_1 \in h_1, \ \mu_2 \in h_2),$$

where $c_{\text{cnn}}$ is a constant independent of depth, kernel size $k_h$, channel width $C_h$, and spatial resolution $N_h$.

**Depth scaling and width-invariant leading term.** Averaging over layers $\ell = 1,\ldots,L$, we obtain

$$\frac{1}{L} \sum_{\ell=1}^{L} \mathbb{E}\left[(\Delta z^{(\ell)})^2\right] = \Theta(\eta^2) \cdot \frac{1}{L} \sum_{\ell=1}^{L} \sum_{h_1=1}^{\ell} \sum_{h_2=1}^{\ell} \min\{h_1,h_2\}.$$

Using the identity

$$\sum_{h_1=1}^{\ell} \sum_{h_2=1}^{\ell} \min\{h_1,h_2\} = \frac{\ell(\ell+1)(2\ell+1)}{6} = \Theta(\ell^3),$$

we deduce

$$\frac{1}{L}\sum_{\ell=1}^{L}\mathbb{E}\Big[(\Delta z^{(\ell)})^2\Big] = \Theta\big(\eta^2 L^3\big).$$

Normalizing the "stable step size" by requiring $\frac{1}{L}\sum_{\ell}\mathbb{E}[(\Delta z^{(\ell)})^2] \asymp 1$ yields

$$\boxed{\eta^{\star}(L) = \kappa\, L^{-3/2}}$$

where $\kappa$ depends only on the fixed-point constant of ReLU+He initialization, and is independent of channel widths $C_\ell$, kernel sizes $k_\ell$, and spatial resolutions $N_\ell$.

**Finite-width and boundary corrections.** In more general CNNs (e.g. with zero padding, mild channel heterogeneity, or finite mini-batches), the deviation from exact unit-wise equality contributes only subleading corrections, which can be summarized as

$$\boxed{\eta^{\star}(L, \{C_\ell, N_\ell, k_\ell\}) = \kappa\, L^{-3/2}\Big(1 + O\Big(\underbrace{\max_{\ell}\tfrac{1}{C_\ell - 1}}_{\text{width term}} + \underbrace{\max_{\ell}\tfrac{s_\ell}{N_\ell}}_{\text{boundary term}} + \underbrace{\tfrac{1}{B}}_{\text{batch variance}}\Big)\Big).}$$

Thus, channel width $C_{\ell-1}$ only enters through $O(1/C_{\ell-1})$ corrections, leaving the $-3/2$ depth exponent intact. Likewise, common zero padding produces $O(s_\ell/N_\ell)$ boundary effects, which vanish as feature maps grow.

This completes the proof of Theorems 1 and 2 in 1D CNN.

**2D case (differences only; proof by analogy).** Replace the 1D spatial index set by a 2D grid $\Lambda_h = \{1, \ldots, H_h\} \times \{1, \ldots, W_h\}$ (so $N_h = H_h W_h$). Let the kernel offset set be $\mathcal{K}_h \subset \mathbb{Z}^2$ (arbitrary shape), with cardinality $k_h := |\mathcal{K}_h|$. Keep stride $= 1$, circular padding, ReLU, and He fan-in with $\mathrm{Var}(W_{j,i,\Delta}^{(h)}) = 2/(C_{h-1} k_h)$.

*Layerwise conditional expectation invariance (2D).* The proof is identical to Lemma 1 after reindexing $\sum_{p\in\Lambda_h}\sum_{\Delta\in\mathcal{K}_h}$ on the torus to $\sum_{u\in\Lambda_{h-1}}$: when $(p, \Delta)$ jointly range, every previous-layer site $u$ is visited exactly $k_h$ times, which cancels the fan-in factor $1/k_h$; also $N_h = N_{h-1}$ under stride $= 1$ with circular padding. Hence

$$\mathbb{E}\Big[T_h(\mu_1, \mu_2) \mid z^{(h-1)}\Big] = T_{h-1}(\mu_1, \mu_2), \qquad \mathbb{E}\, T_h(\mu_1, \mu_2) = \mathbb{E}\, T_{h-1}(\mu_1, \mu_2).$$

*Top-layer decomposition and magnitude of A.* As in Lemma 2, with $N_{L+1}$ replaced by $H_{L+1}W_{L+1}$, the weak-dependence estimate yields

$$\mathbb{E}\big[S_{L+1}(\mu_1, \mu_2)\big] = O\big((C_{L+1}H_{L+1}W_{L+1})^{-1}\big), \qquad \mathbb{E}\big[T_{L+1}(\mu_1, \mu_2)\big] = O(1),$$

so $A_{\mathrm{cnn}}^{(\ell)} = O\big((C_{L+1}H_{L+1}W_{L+1})^{-1}\big)$ remains negligible.

*Unit averaging and overlap counting.* With homogeneity (channel i.i.d., spatial stationarity, stride $= 1$, circular padding) and the above invariance, the 2D analogue of the unit–average relation gives

$$\mathbb{E}\big[T_\ell(\mu_1, \mu_2)^2\big] = c_{\mathrm{cnn}} \cdot \min\{h_1, h_2\} \quad (\mu_1 \in h_1,\ \mu_2 \in h_2),$$

where $c_{\mathrm{cnn}}$ is independent of $\{C_h\}$, $\{\mathcal{K}_h\}$, and $(H_h, W_h)$. Consequently,

$$\frac{1}{L}\sum_{\ell=1}^{L}\mathbb{E}\big[(\Delta z^{(\ell)})^2\big] = \Theta(\eta^2) \cdot \frac{1}{L}\sum_{\ell=1}^{L}\sum_{h_1, h_2 \le \ell}\min\{h_1, h_2\} = \Theta(\eta^2 L^3),$$

and the stable scale satisfies

$$\boxed{\eta^{\star}(L) = \kappa\, L^{-3/2}\,,}$$

with $\kappa$ depending only on the ReLU+He fixed point and independent of $\{C_h\}$, $\{\mathcal{K}_h\}$, and $(H_h, W_h)$.

**Remark** (2D corrections). *With non-circular padding (e.g., zero padding), boundary visits are nonuniform. Let the maximal axial spans of the kernel be $s_{h,h} := \max_{\Delta \in \mathcal{K}_h} |\Delta_h|$ and $s_{h,w} := \max_{\Delta \in \mathcal{K}_h} |\Delta_w|$. Then*

$$\mathbb{E}\Big[T_h \mid z^{(h-1)}\Big] = T_{h-1} + O\Big(\frac{s_{h,h}}{H_{h-1}} + \frac{s_{h,w}}{W_{h-1}}\Big).$$

*Together with finite–channel and mini-batch effects, we obtain the unified 2D correction:*

$$\boxed{\eta^\star\big(L; \{C_h, H_h, W_h, \mathcal{K}_h, B\}\big) = \kappa\, L^{-3/2}\left(1 + O\Big(\max_h \frac{1}{C_{h-1}}\Big) + O\Big(\max_h \frac{s_{h,h}}{H_h} + \frac{s_{h,w}}{W_h}\Big) + O\Big(\frac{1}{B}\Big)\right),}$$

*which does not change the depth exponent $-3/2$.*

This completes the proof of Theorems 1 and 2 in 2D CNN.

## C   PROOF OF SCALING LAW FOR RESNETS

**Lemma 3** (Layerwise scaling recursion for one-layer residual blocks). *Consider a homogeneous residual network whose $\ell$-th block is the one-layer (MLP-like) residual map*

$$z^{(\ell)} = W^{(\ell)} \sigma\big(z^{(\ell-1)}\big) + z^{(\ell-1)},$$

*with identity skip, ReLU activation, and no normalization. Assume the weights are independent and zero-mean with fan-in variance $\text{Var}\big(W_{ik}^{(\ell)}\big) = \frac{c}{Kn}$, and $\mathbb{E}[\sigma'(u)^2] = \frac{1}{2}$. For any two parameter directions $\mu_1, \mu_2$, define*

$$T_\ell(\mu_1, \mu_2) := \frac{1}{n} \sum_{i=1}^n \partial_{\mu_1} z_i^{(\ell)}\, \partial_{\mu_2} z_i^{(\ell)}.$$

*Then, for every block (layer) $\ell$,*

$$\boxed{\mathbb{E}\Big[T_\ell(\mu_1, \mu_2) \,\Big|\, z^{(\ell-1)}\Big] = \Big(1 + \frac{c}{2K}\Big) T_{\ell-1}(\mu_1, \mu_2).}$$

*Proof.* The residual block forward map is $z^{(\ell)} = W^{(\ell)} \sigma\big(z^{(\ell-1)}\big) + z^{(\ell-1)}$. For any parameter direction $\mu$,

$$\partial_\mu z_i^{(\ell)} = \sum_k W_{ik}^{(\ell)}\, \sigma'\big(u_k^{(\ell)}\big)\, \partial_\mu z_k^{(\ell-1)} + \partial_\mu z_i^{(\ell-1)}.$$

Substitute this into $T_\ell = \frac{1}{n} \sum_i \big(\partial_{\mu_1} z_i^{(\ell)}\big)\big(\partial_{\mu_2} z_i^{(\ell)}\big)$, expand into the three groups (W–W, I–I, and cross W–I), and take conditional expectation over the $\ell$-th layer weights given $z^{(\ell-1)}$:

*(i) Cross terms (W–I):* Every term contains one factor of $W^{(\ell)}$ and vanishes by $\mathbb{E}[W] = 0$.

*(ii) I–I term:*

$$\mathbb{E}[\text{I–I} \mid z^{(\ell-1)}] = \frac{1}{n} \sum_{i=1}^n \partial_{\mu_1} z_i^{(\ell-1)}\, \partial_{\mu_2} z_i^{(\ell-1)} = T_{\ell-1}(\mu_1, \mu_2).$$

*(iii) W–W term:* Only the diagonal $k = k'$ survives by independence,

$$\begin{aligned}
\mathbb{E}[\text{W–W} \mid z^{(\ell-1)}] &= \frac{1}{n} \sum_i \sum_{k,k'} \mathbb{E}\Big[W_{ik}^{(\ell)} W_{ik'}^{(\ell)}\Big]\, \sigma'\big(u_k^{(\ell)}\big)\sigma'\big(u_{k'}^{(\ell)}\big)\, \partial_{\mu_1} z_k^{(\ell-1)}\, \partial_{\mu_2} z_{k'}^{(\ell-1)} \\
&= \frac{1}{n} \sum_i \sum_k \text{Var}\big(W_{ik}^{(\ell)}\big)\, \mathbb{E}\Big[\sigma'\big(u_k^{(\ell)}\big)^2\Big]\, \partial_{\mu_1} z_k^{(\ell-1)}\, \partial_{\mu_2} z_k^{(\ell-1)} \\
&= \text{Var}[W] \sum_k \mathbb{E}\Big[\sigma'\big(u_k^{(\ell)}\big)^2\Big]\, \frac{1}{n} \sum_k \partial_{\mu_1} z_k^{(\ell-1)}\, \partial_{\mu_2} z_k^{(\ell-1)} \\
&= \frac{c}{2K}\, T_{\ell-1}(\mu_1, \mu_2),
\end{aligned}$$

where we used $\mathbb{E}[W_{ik} W_{ik'}] = 0$ for $k \neq k'$, $\text{Var}[W] = \frac{c}{Kn}$, and $\mathbb{E}[\sigma'(u)^2] = \frac{1}{2}$.

Combining (i)–(iii) yields $\mathbb{E}[T_\ell \mid z^{(\ell-1)}] = \big(1 + \frac{c}{2K}\big) T_{\ell-1}$, as claimed. $\qquad\square$

**Lemma 4** (Magnitude form of $B_{\text{res}}^{(\ell)}$). *Under the assumptions of Lemma 3 (homogeneous ResNet with identity skips, ReLU, independent zero-mean weights with fan-in variance $\text{Var}[W] = c/(Kn)$, and $\mathbb{E}[\sigma'(u)^2] = \frac{1}{2}$), we have*

$$B_{\text{res}}^{(\ell)} = \Theta\left(\mathbb{E}\left[\frac{1}{n}\sum_{\mu_1,\mu_2\leq\ell}\eta_{\mu_1}\eta_{\mu_2}\frac{1}{n^2}\sum_{j_1,j_2=1}^{n}\partial_{\mu_1}z_{j_1;\alpha}^{(\ell)}\,\partial_{\mu_2}z_{j_1;\alpha}^{(\ell)}\,\partial_{\mu_1}z_{j_2;\alpha}^{(\ell)}\,\partial_{\mu_2}z_{j_2;\alpha}^{(\ell)}\right]\right).$$

*Equivalently, the right-hand side is proportional to $\mathbb{E}\left[\sum_{\mu_1,\mu_2\leq\ell}\eta_{\mu_1}\eta_{\mu_2}\,T_\ell(\mu_1,\mu_2)^2\right]$, where $T_\ell(\mu_1,\mu_2) := \frac{1}{n}\sum_{i=1}^{n}\partial_{\mu_1}z_i^{(\ell)}\,\partial_{\mu_2}z_i^{(\ell)}$.*

*Proof sketch.* Using the same top-layer decomposition as in Lemma 2 and diagonal dominance under weak inter-unit dependence, the output-layer second-order term reduces to the stated $\Theta(T_\ell^2)$ magnitude; subleading off-diagonal contributions are $O(1/n)$ and are absorbed into the $\Theta(\cdot)$ notation. □

**Comparison factor and $O(1)$ bounds (ResNet vs. MLP).** By the layerwise scaling recursion of Lemma 3, iterating from layer 0 to $\ell$ yields a layer-dependent factor

$$r_\ell := \left(1 + \tfrac{c}{2K}\right)^\ell$$

so that, at the same depth $\ell$,

$$B_{\text{res}}^{(\ell)} = r_\ell \cdot B_{\text{MLP}}^{(\ell)} \quad \text{(under the same top-layer reduction and normalization).}$$

Since $0 \leq \ell \leq K$,

$$1 = \left(1 + \tfrac{c}{2K}\right)^0 \leq r_\ell \leq \left(1 + \tfrac{c}{2K}\right)^K \leq e^{c/2},$$

and hence, symmetrically,

$$e^{-c/2}\,B_{\text{MLP}}^{(\ell)} \leq B_{\text{res}}^{(\ell)} \leq e^{c/2}\,B_{\text{MLP}}^{(\ell)},$$

showing that even across $\ell \leq K$ residual blocks the $B$-term varies only by an $O(1)$ multiplicative constant, with no exponential blow-up or vanishing in depth.

**Corollary** (Depth scaling for homogeneous ResNets (minimal depth)). *Let $L$ denote the minimal depth, i.e., each residual block counts as one layer (regardless of its internal linear/convolutional sublayers). Under the assumptions of Lemma 3 (identity skips, ReLU, independent zero-mean fan-in initialization), combining the top-layer decomposition in Lemma 2 with the layerwise invariance and overlap-counting argument in Appendix B (Lemma 1), we obtain for every $\ell \leq L$:*

$$\mathbb{E}\left[(\Delta z^{(\ell)})^2\right] = \Theta(\eta^2\,\ell^3).$$

*Averaging over layers $\ell = 1,\ldots,L$ yields*

$$\frac{1}{L}\sum_{\ell=1}^{L}\mathbb{E}\left[(\Delta z^{(\ell)})^2\right] = \Theta(\eta^2\,L^3).$$

*Imposing the stable step-size condition $\frac{1}{L}\sum_{\ell=1}^{L}\mathbb{E}\left[(\Delta z^{(\ell)})^2\right] = \Theta(1)$ gives*

$$\eta^\star(L) = \Theta(L^{-3/2})$$

*Proof sketch.* *Lemma 3 shows a layerwise scaling of the quadratic sensitivities by $(1 + \frac{c}{2K})$, which contributes only an $O(1)$ factor uniformly in $\ell$ and is absorbed into the $\Theta(\cdot)$ notation. The remaining steps (top-layer reduction and overlap counting) follow exactly as in Appendix B.*

### C.1 EXTENSIONS

We use the *minimal depth* convention: each residual block counts as one layer. Let $L$ be the minimal depth and $K$ the number of residual blocks.

### C.1.1 DEEPER RESIDUAL BRANCHES

**Corollary** (Shallow residual branches: $m = o(L)$). *Consider the $\ell$-th residual block whose branch contains $m$ repetitions of "ReLU $\to$ linear/conv" transformations, followed by a final merge via $W^{(\text{res})}$ and identity skip:*

$$z^{(\ell)} = W^{(\text{res})} \sigma\big(\cdots\sigma(W^{(r_1)}\sigma(z^{(\ell-1)}))\cdots\big) + z^{(\ell-1)}.$$

*Assume fan-in type initialization scaled by the number of blocks,* $\text{Var}[W] = c/(K \cdot \text{fan-in})$, *and no normalization. If $m = o(L)$ as $L \to \infty$, then for every $\ell \leq L$,*

$$\mathbb{E}\big[(\Delta z^{(\ell)})^2\big] = \Theta(\eta^2\,\ell^3), \qquad \frac{1}{L}\sum_{\ell=1}^{L}\mathbb{E}\big[(\Delta z^{(\ell)})^2\big] = \Theta(\eta^2\,L^3),$$

*and hence the stable step size scales as*

$$\boxed{\eta^\star(L) = \Theta\big(L^{-3/2}\big)}$$

Proof sketch. *Each intermediate "ReLU $\to$ weight" inside the branch contributes a W–W increment proportional to $\text{Var}[W] = O(1/K)$, so the total branch-level increment is $m \cdot O(1/K) = o(1)$ and is absorbed into the $\Theta(\cdot)$ constants. The only leading-order change comes from the final $W^{(\text{res})}$ merged with the identity skip, whose layerwise scaling is controlled by Lemma 3. The top-layer reduction and overlap counting then proceed exactly as in Appendix B.*

### C.1.2 RESIDUAL-BLOCK STRUCTURAL EXTENSION: ALLOWING CONVOLUTIONS IN THE BRANCH

**Corollary** (Residual branches with 1D convolutions). *In the setting of Corollary C.1.1, allow one or a few 1D/2D convolutional layers inside the branch (stride = 1, circular padding or effectively boundary-free), with fan-in scaled He-type initialization (again multiplied by $1/K$ at the block level). If the total number of branch layers still satisfies $m = o(L)$, then the depth scaling remains*

$$\mathbb{E}\big[(\Delta z^{(\ell)})^2\big] = \Theta(\eta^2\,\ell^3), \qquad \eta^\star(L) = \Theta\big(L^{-3/2}\big).$$

Proof sketch. *The proof follows the same logic as Corollary C.1.1: each convolutional layer inside the branch carries the same $O(1/K)$ variance factor and its W–W increment is therefore $O(1/K)$; summing over $m = o(L)$ branch layers yields $O(m/K) = o(1)$ per block, which is absorbed into the $\Theta(\cdot)$ constants. CNN-specific boundary/width/batch corrections (e.g., $O(1/C_\ell)$, $O(s_\ell/N_\ell)$, $O(1/B)$) are lower-order and do not affect the $\ell^3$ and $L^{-3/2}$ Theta-level conclusions. The leading term is again governed by the merge through $W^{(\text{res})}$ and the identity skip, after which the top-layer reduction and overlap counting proceed as in Appendix B.*

This completes the proof of Theorem 3.

## D MORE EXPERIMENTS

### D.1 CNN: ADDITIONAL EXPERIMENTS

**GELU-specific settings.** We use the same homogeneous 2D convolutional blocks (stride 1, circular padding), optimizer (SGD without momentum, batch size 128), and one-epoch protocol as in Sec. 4.1. The only differences are: (i) the activation is GELU; (ii) we adjust He fan-in initialization by multiplying the variance by $\sqrt{2}$ to align the activation fixed point with ReLU.[5] Depth $L$ counts `conv + nonlinearity` blocks; the classifier is global pooling followed by a linear head. A complete panel for CIFAR-10 with GELU is shown in Fig. 3.

---

[5]This alignment keeps pre-activation variance approximately depth-stationary, isolating the activation effect on the exponent; see (Chen, 2024; Jelassi et al., 2023).

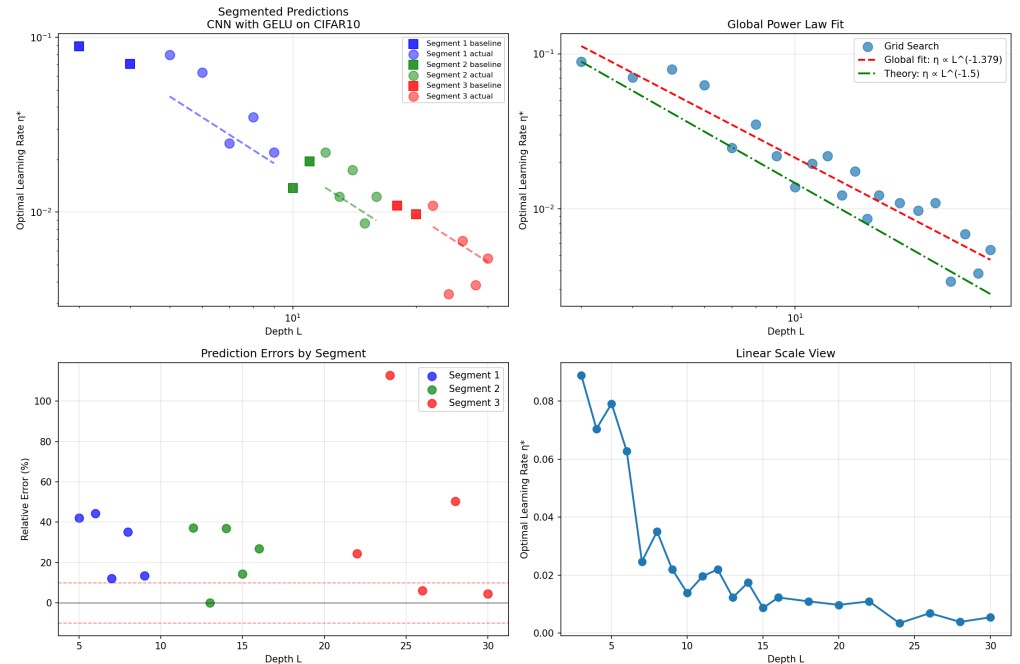

Figure 3: **CNN on CIFAR-10 (GELU): full panel.** Top-left: segmented predictions using two anchor depths per segment (A/B/C). Top-right: global power-law fit of $\eta^\star$ vs. $L$ with slope $\hat{\alpha} \approx -1.38$ (red dashed), shown against a reference line (green dash-dotted). Bottom-left: relative errors by segment, with larger deviations near segment boundaries and at the largest depths. Bottom-right: linear-scale view showing the rapid decay of the maximal-update learning rate $\eta^\star$ with depth.

Across this additional CNN setting (CIFAR-10 with GELU), the maximal-update learning rate follows a clear depth power law. The global fit yields $\hat{\alpha} \approx -1.38$; segmented two-anchor fits extrapolate well within segments, while errors increase near segment boundaries and for the deepest models. See Fig. 3 for the full panel.

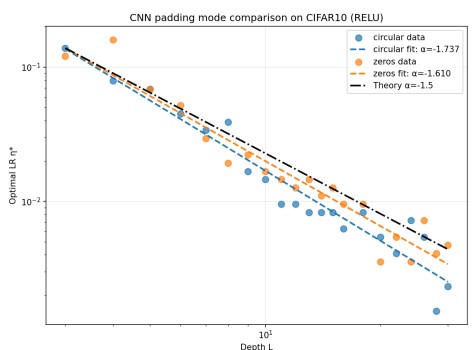

Figure 4: CNN padding comparison on CIFAR-10 (ReLU).

**Padding ablation (circular vs. zero).** We compare circular and zero padding under identical CNN settings on CIFAR-10 (ReLU). Both padding modes follow essentially the same depth–learning-rate power law with exponents close to the $L^{-3/2}$ prediction; differences are mainly a small vertical shift on the log scale (i.e., a prefactor change) rather than a slope change. Hence, padding has a minor effect on the scaling law, and zero padding is a practical default in engineering.

Beyond CIFAR-10, we also evaluate CNNs on CIFAR-100. Across this additional CNN setting (CIFAR-100 with ReLU), the maximal-update learning rate follows a clear depth power law. The global fit yields $\hat{\alpha} \approx -1.392$, consistent with the $-3/2$ prediction. Segmented two-anchor fits extrapolate well within segments, while errors increase near segment boundaries and for the deepest models. See Fig. 5 for the full panel.

### D.2 RESNET: ADDITIONAL EXPERIMENTS (BATCHNORM/DROPOUT)

We investigate whether standard regularizers used in practice—batch normalization (BN) and dropout—modify the depth–learning-rate law. Specifically, we replicate our ResNet study under

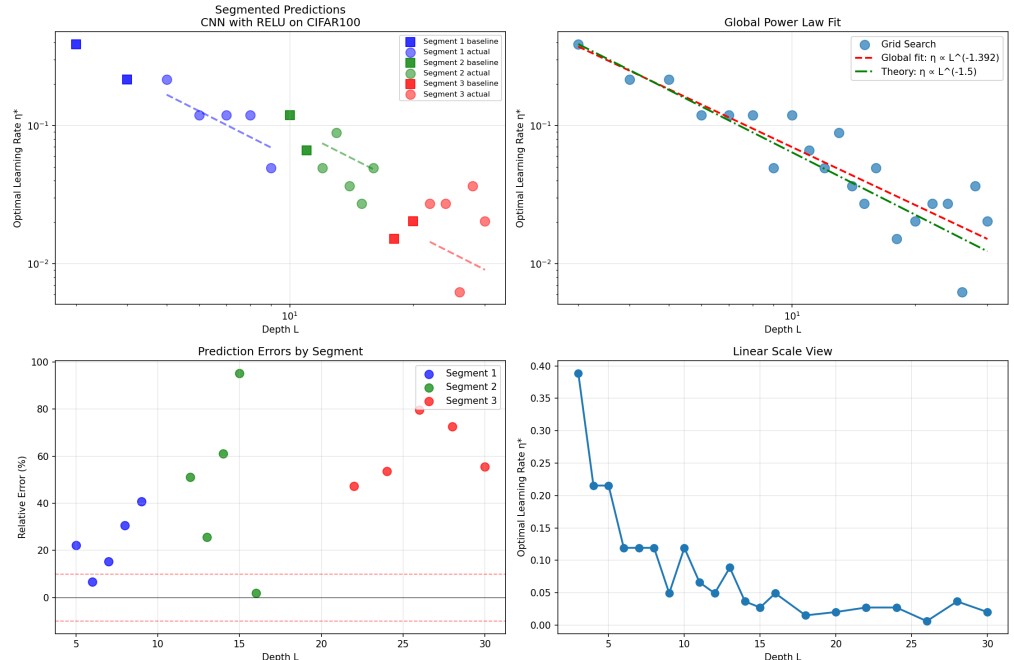

Figure 5: **CNN on CIFAR-100 (ReLU): full panel.** Top-left: segmented predictions using two anchor depths per segment (A/B/C). Top-right: global power-law fit of $\eta^\star$ vs. $L$ with slope $\hat{\alpha} \approx -1.392$ (red dashed), closely tracking the $L^{-3/2}$ reference (green dash-dotted). Bottom-left: relative errors by segment, with larger deviations near segment boundaries and at the largest depths. Bottom-right: linear-scale view showing the rapid decay of the maximal-update learning rate $\eta^\star$ with depth.

four variants: (i) BN only, (ii) dropout only, (iii) BN+dropout, and (iv) none. The "none" variant matches our main ResNet setting; we report it only on CIFAR-100 for completeness. All experiments follow the protocol in Sec. 4.1: we identify the maximal-update learning rate $\eta^\star$ after one epoch on a logarithmic grid and evaluate segmented zero-shot depth transfer. The goal is to test whether these regularizers change the exponent $\alpha$ in $\eta^\star \propto L^{-\alpha}$ or primarily shift the prefactor $\kappa$ by altering gradient scale and noise statistics.

Across ResNet variants, log–log fits yield a stable power law $\eta^\star \propto L^{-\alpha}$ with $|\alpha| \approx 1.5$–1.6; equivalently, $\log(1/\eta^\star)$ increases approximately linearly with $\log L$. Under Dropout, both CIFAR-10 and CIFAR-100 give $|\alpha| \approx 1.56$; with BatchNorm, the global slopes are $|\alpha| \approx 1.869$ and $1.399$ (mean $1.634$); with BatchNorm+Dropout, $|\alpha| \approx 1.56$. These differences are small and consistent with expected estimation noise (finite-width, padding/boundary effects, and the one-epoch proxy), indicating that the depth–learning-rate rule is robust to these regularizers.

### D.3    IMAGENET: ADDITIONAL SCALING RESULTS

We repeat the maximal-update LR search on ImageNet with the same logarithmic grid as in Sec. 4.1. For each depth $L$, we train for *one full epoch* (a complete pass over the ImageNet training set) and record $\eta^\star$ at the end of the epoch. Other settings mirror Sec. 4.1 (SGD without momentum, He fan-in); the batch size follows the standard ImageNet recipe and is held constant across depths.

Across ImageNet-scale runs, $\eta^\star$ decays predictably with depth: the CNN yields $\hat{\alpha} \approx -1.329$ (Fig. 8), the ResNet with dropout yields $\hat{\alpha} \approx -1.663$ (Fig. 9), and the ResNet without dropout yields $\hat{\alpha} \approx -1.567$ (Fig. 10). These values are consistent with the $L^{-3/2}$ rule and align with the CIFAR results.

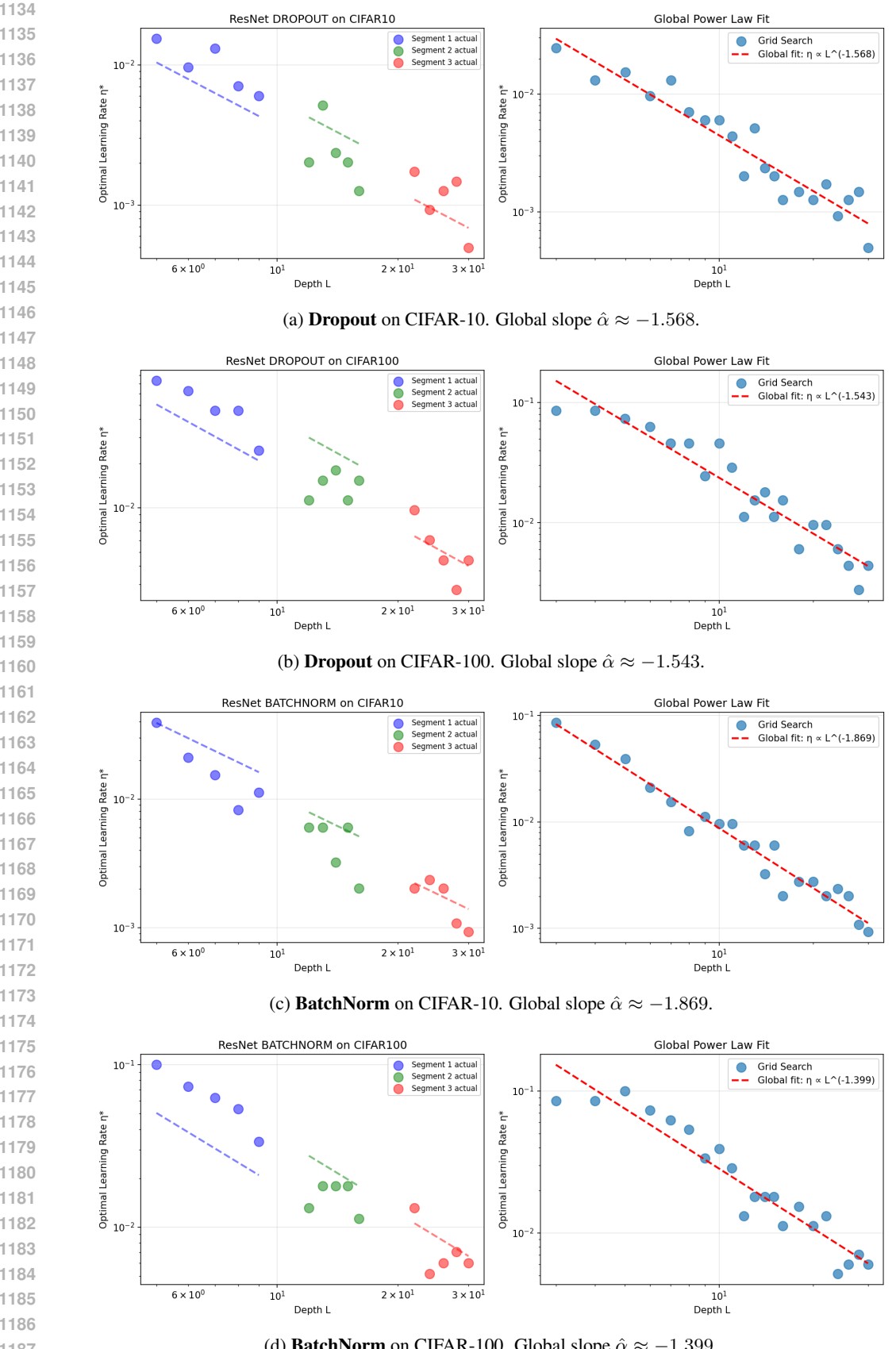

(a) **Dropout** on CIFAR-10. Global slope $\hat{\alpha} \approx -1.568$.

(b) **Dropout** on CIFAR-100. Global slope $\hat{\alpha} \approx -1.543$.

(c) **BatchNorm** on CIFAR-10. Global slope $\hat{\alpha} \approx -1.869$.

(d) **BatchNorm** on CIFAR-100. Global slope $\hat{\alpha} \approx -1.399$.

Figure 6: **ResNet variants: Dropout and BatchNorm.** Each row shows a full panel (segmented predictions + global power-law fit) for the specified variant and dataset.

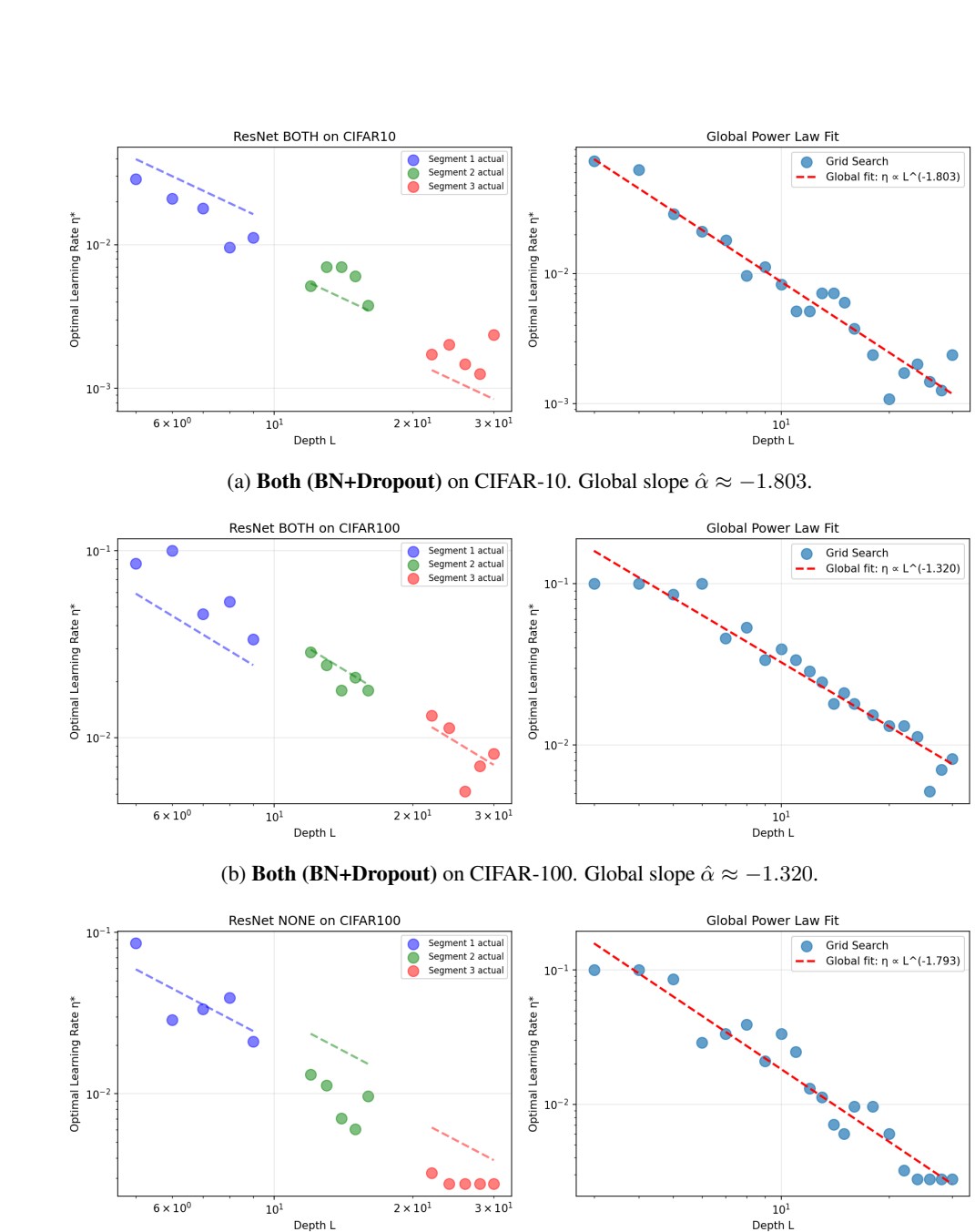

(a) **Both (BN+Dropout)** on CIFAR-10. Global slope $\hat{\alpha} \approx -1.803$.

(b) **Both (BN+Dropout)** on CIFAR-100. Global slope $\hat{\alpha} \approx -1.320$.

(c) **None** (no BN/Dropout) on CIFAR-100. Global slope $\hat{\alpha} \approx -1.793$.

Figure 7: **ResNet variants: None and Both.** Panels (top to bottom): Both (BN+Dropout) on CIFAR-10; Both (BN+Dropout) on CIFAR-100; None (no BN/Dropout) on CIFAR-100.

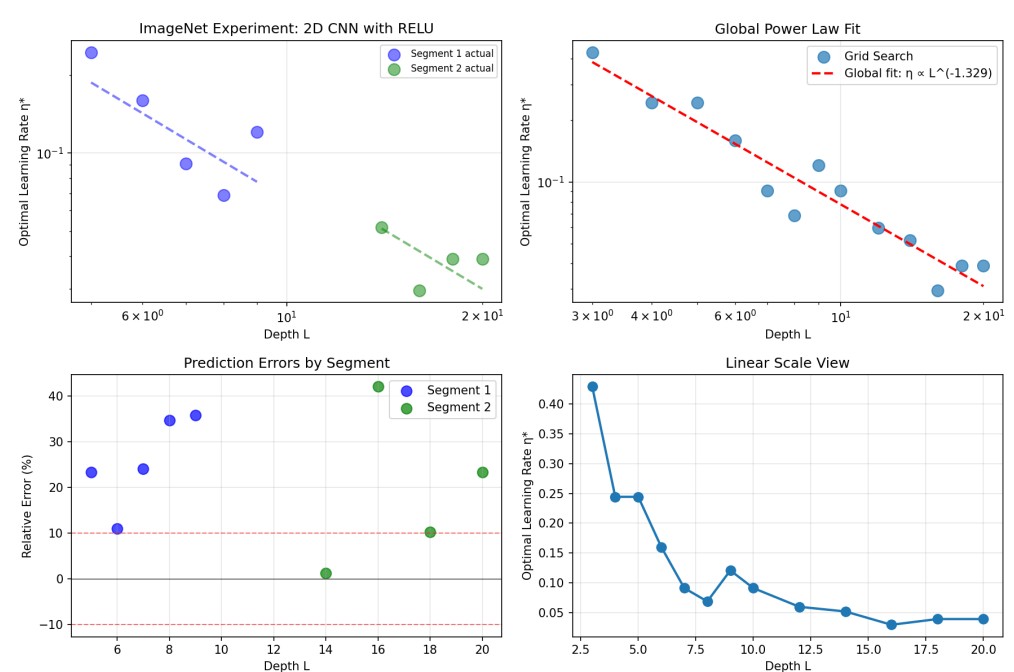

Figure 8: **ImageNet: 2D CNN (ReLU), full panel.** Top-left: segmented two-anchor predictions (two segments). Top-right: global log–log fit of $\eta^\star$ vs. $L$ with slope $\hat{\alpha} \approx -1.329$ (red dashed). Bottom-left: segment-wise relative errors. Bottom-right: linear-scale view of $\eta^\star$ vs. depth.

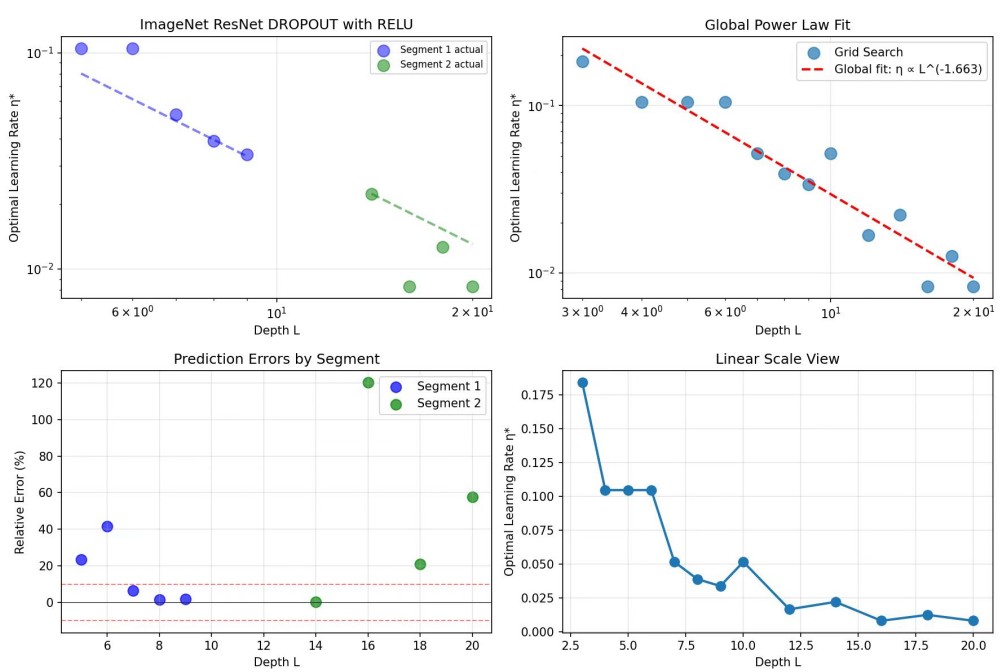

Figure 9: **ImageNet: ResNet (ReLU) with Dropout, full panel.** Top-left: segmented two-anchor predictions (two segments). Top-right: global log–log fit of $\eta^\star$ vs. $L$ with slope $\hat{\alpha} \approx -1.663$ (red dashed). Bottom-left: segment-wise relative errors. Bottom-right: linear-scale view of $\eta^\star$ vs. depth.

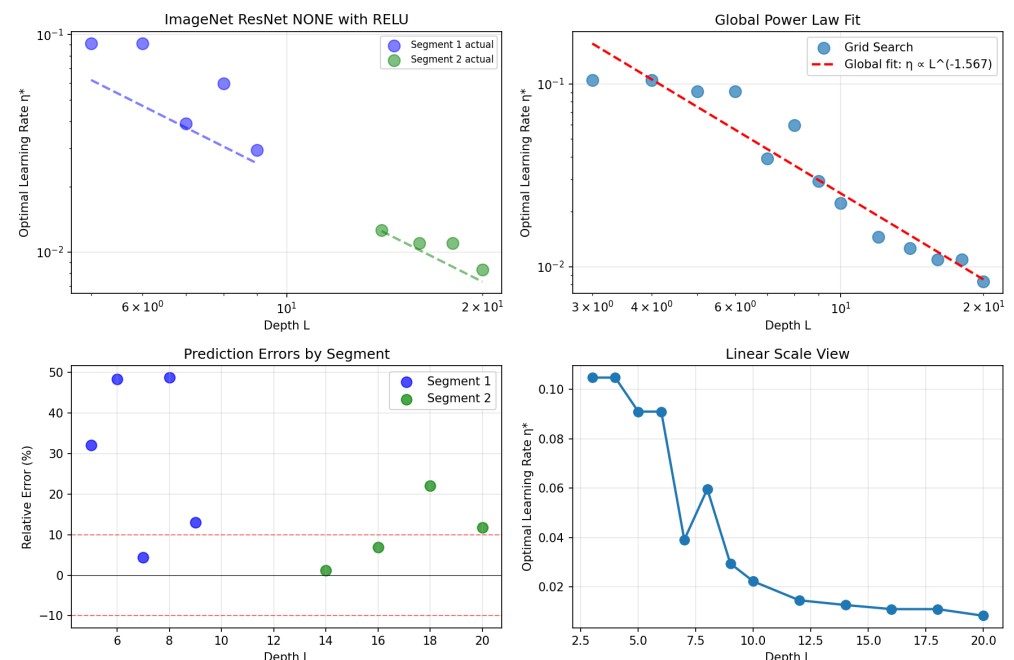

Figure 10: **ImageNet: ResNet (ReLU)** *without* **BN/Dropout, full panel.** Top-left: segmented two-anchor predictions. Top-right: global log–log fit of $\eta^\star$ vs. $L$ (red dashed). Bottom-left: segment-wise relative errors. Bottom-right: linear-scale view of $\eta^\star$ vs. depth.

# E  WHY GELU EXHIBITS A SLIGHTLY STEEPER DEPTH–LR EXPONENT THAN RELU

Empirically, the fitted depth–learning-rate exponent for GELU is marginally more negative than for ReLU (e.g., $-1.40$ vs. $-1.35$). This small gap can be attributed to two effects:

**Activation–derivative statistics.** With fan-in initialization adjusted to keep $z \sim \mathcal{N}(0,1)$ (as in Sec. D.1), ReLU satisfies

$$\mathbb{E}[\sigma'(z)^2] = 0.5,$$

whereas GELU $\phi(x) = x\Phi(x)$ yields

$$\mathbb{E}[\phi'(z)^2] \approx 0.456.$$

The resulting expected Jacobian factor per layer is therefore slightly smaller for GELU ($\chi = 2\,\mathbb{E}[\phi'(z)^2] \approx 0.912$ vs. 1 for ReLU), which lowers the effective constant in the depth–LR scaling and, over a finite depth range, manifests as a slightly more negative fitted exponent in log–log regression.

**Finite-depth/width corrections.** Small variance drifts across layers alter $\mathbb{E}[\phi'(z)^2]$ along depth; GELU is more sensitive to such drifts because $\phi'$ depends smoothly on $z$. This induces a mild, depth-dependent attenuation of the effective step size in deeper layers, which—when regressed as a single power law—manifests as a slightly more negative fitted exponent.

# F  ON THE LOSS: CROSS-ENTROPY VS. MSE IN THE DERIVATION

Our theoretical derivation uses MSE for analytic convenience in the one-step maximal-update analysis, whereas all experiments use multi-class cross-entropy (CE). This mismatch does not affect the depth exponent.

At initialization, logits are near zero and $\mathrm{softmax}(z)$ is close to uniform. For one-hot targets $y$ with $C$ classes, the CE logit gradient is

$$g \;=\; p - y, \qquad p = \mathrm{softmax}(z),$$

so that

$$\|g\|_2^2 \;=\; \left(1 - \tfrac{1}{C}\right)^2 + (C-1)\left(\tfrac{1}{C}\right)^2 \;=\; 1 - \tfrac{1}{C} \;=\; O(1).$$

Hence CE provides $O(1)$-scale per-sample gradients in early training, the regime in which we identify $\eta^\star$. Under our He/$\mu$P parameterization, the depth dependence of $\eta^\star(L)$ is governed by architecture (Jacobian products), so swapping MSE for CE only rescales the overall prefactor $\kappa$ and does *not* change the power-law exponent. Empirically, CE and MSE produce nearly parallel $\log \eta^\star$–$\log L$ fits with the same slope, differing by a vertical shift (prefactor).

