# OpenReview forum: "Arithmetic-Mean $\mu$P for Modern Architectures: A Unified Learning-Rate Scale for CNNs and ResNets"
_ICLR.cc/2026/Conference — Submitted to ICLR 2026_

### Official Review · Reviewer_YPVT · 2025-10-22

**Soundness:** 1
**Presentation:** 2
**Contribution:** 1
**Rating:** 2
**Confidence:** 4

**Summary:**

The paper studies scaling laws for the learning rate for convolutional neural networks (CNNs) and residual networks (ResNets). The paper introduces a new condition for stability, which is the average variance of the pre-activation gradient updates, where the average is taken over the depth of the network. This condition implies a scaling for the learning rate, which scales as $L^{-3/2}$ both for CNNs and for ResNets. Experiments corroborate the findings.

**Strengths:**

Deriving scaling laws is essential to modern deep learning. The authors study realistic architectures, including for instance the effect of padding in CNNs.

**Weaknesses:**

W1: there is already a strong literature on learning rate scaling for ResNets (and CNNs) that the authors do not engage with, e.g.,

The Feature Speed Formula: a flexible approach to scale hyper-parameters of deep neural networks, Chizat and Netrapalli, NeurIPS 2024. Tthis paper is cited by the authors, but they do not compare the results, although this reference already gives formulas for ResNets.

Tensor Programs VI: Feature Learning in Infinite-Depth Neural Networks, Yang et al., ICLR 2024. Gives formulas for ResNets.

Tensor Programs V: Tuning Large Neural Networks via Zero-Shot Hyperparameter Transfer, Yang et al., NeurIPS 2021. Mentions that the results can be extended to CNNs (see remark page 47).

The authors should explain how their work connects to the literature.

W2: for ResNets, it is known (cf literature above, or also Marion et al., Scaling ResNets in the Large-depth Regime, JMLR 2025) that the residual branch should be scaled in order to avoid instability. This scaling interacts with the scaling of the learning rate, which is then different from the MLP case. In the present paper, this scaling is not considered. This questions the validity of the learning rate scaling derived in the paper (since the forward pass already leads to instabilities).

W3: the condition on the average variance that is introduced in the paper relies on the idea that the scale of the updates is heterogeneous along the depth of the network. However, most of the literature precisely tries to avoid such heterogeneity, by scaling appropriately both the forward and backward pass (see again references above). So the authors would need to precisely justify why they consider the heterogeneous case. Furthermore, if the updates are heterogeneous, why don’t the authors consider a learning rate that varies depending on the layers?

**Questions:**

See weaknesses.

---

### Official Review · Reviewer_QUSQ · 2025-10-31

**Soundness:** 2
**Presentation:** 1
**Contribution:** 2
**Rating:** 2
**Confidence:** 2

**Summary:**

In deep learning training, the so-called "mu parametrization" (muP, Yang et al 2022) is a scaling of the learning rate as a function of layers' width, designed such that the variance of the first gradient descent update of preactivations, for each layer $\ell$, $S_\ell$, is equal to 1.
The paper argues that enforcing all layerwise variances to be equal to 1 is too restrictive for "heterogeneous networks" such as resNets and CNNs, which have layer that differ in nature (fully connected vs residual or convolutional layers).
The contribution of the paper is to instead enforce that **the average of all $S_\ell$'s for the first update is equal to 1**.
The authors prove that the learning rate that enforces this scales like $L^{-3/2}$ for both resnets and CNNs, a result similar to existing muP results for standard MLPs. Deriving this scaling law allows for fast prototyping and learning rate tuning on small network, than transferring observations to wider and deeper networks.

**Strengths:**

Deriving scaling laws has been a major progress in hyperparameter tuning these last years, allowing to test small architectures and then scale them gracefully. To my knowledge, studies on the learning rate like muP were so far limited to MLPs: extending this work to other architectures is of interest.

**Weaknesses:**

The paper's writing and organization is subpar, which makes it very hard to follow. Some quantities do not seem to be defined (where is the definition of $N_{\ell,r}$, of $\Delta_r$?)
- The abstract mention mathematical quantities that are not defined; while $W$ and fan_in can be inferred, $N$ and $k$ are unknown to the reader at this stage.
- For CNNs, many quantities are not defined ()
- I strongly suggest moving content from the related works section into the introduction.
Indeed, the latter is hard to grasp for a non expert of the domain, as it refers to many concepts not introduced yet (residual accumulation, spatial-channel coupling, minimal effective depth, one-step preactivation update); it is only the related works section that provides the necessary information.
For example, at the introduction stage, the claim "enforcing identical per layer update magnitude is overly restrictive for such heterogeneous networks" does not seem justified; it also remains unclear why zero or circular padding affects variances (sentences such as "visits are uniform and the recursion mirror the FC case" L67 remain cryptic: what are visits, which recursion are the authors thinking about?).
- The part about CNNs is especially unclear: the authors should detail, at least lightly, what are  channel heteroscedasticity, torus indexing, kernel offset set, the axial-half span, etc.
- Still organization-wise, it would make sense to introduce strategies about weight initialization (section 2.3) *before* muP strategies (that requires mean field intialization) in section 2.2.

The paper would require a major rewrite in my opinion.

**Questions:**

- The authors state that He initialization preserves the magnitude of gradients across layers: do they have a reference for that (in particular if network width varies)?
- L205 the authors claim that controlling the average of $S_\ell$'s yields a control over all $S_\ell$, but this does not seem true, as the smallest $S_\ell$ can approach 0 arbitrarily.
- Can the authors define the spatial index set formally? since the paper is focused on convolutions, this would be very useful. What's the kernel offset set? What is the kernel axial span?
- What do the authors mean by $N_\ell = N_{\ell - 1}$ "within a block", followed by "$N_\ell$ is allowed to vary with $\ell$" ?
- The paper refers to "homogeneous cases" as, I believe, cases where all variances per layer are roughly equal. In the literature, homogenous networks are rather relu network, that present some equivariance properties to weight and bias scaling.

Minor remarks:
- All equations should be numbered to facilitate discussion
- The paper uses "Jelassi et al (Jelassi 2023) show that"; this is incorrect. One should use the \citet command of natbib, to produce content that reads: "Jelassi et al (2023) show that". There are several occurrences of this issue.
- the paper uses both fan_in, $n_in$ and $n$ to refer to the number of inputs to the layer in equations; I suggest using systematically $n_in$

---

### Official Review · Reviewer_g7Ee · 2025-11-03

**Soundness:** 3
**Presentation:** 2
**Contribution:** 2
**Rating:** 4
**Confidence:** 3

**Summary:**

The paper studies the scaling of the optimal learning rate $\\eta^*$ with network depth $L$ for convolutional and residual networks. The main result is the scaling law

$$\\eta^* \\propto L^{-3/2}.$$

(appearing earlier for MLPs in Jelassi et al., 2023). This law is established theoretically in theorems 1-3 and verified by a series of experiments on CIFAR 10.

**Strengths:**

The paper addresses a reasonable and important point of scaling the learning rate of gradient descent with network depth. The obtained scaling law, while known previously for MLPs, is established in the paper for the more practical and modern CNN and Resnet architectures. The scaling law is carefully verified experimentally, and the results agree well with the theoretical predictions. The paper is definitely sound and has a practical value.

The paper is on the whole well written (albeit in a relatively informal style, see weaknesses below). Several appendices are provided, covering the proofs of the theorems, additional discussion of the proposed AM $\\mu P$ regime and additional experiments.

**Weaknesses:**

**Limited novelty / significance**

My impression is that the main contribution of the paper is an extension of the previously known $L^{-3/2}$ optimal depth scaling of learning rates (Jelassi et al., 2023) to CNNs and Resnets (in Theorems 1-3). As the authors themselves point out, the scaling was known for MLPs. The present work takes into account the special aspects of CNNs and Resnets (filter size, number of channels, etc.), but it seems that the overall scaling and the main logic of the derivation are the same, and so the extension is relatively incremental - at least, I don't see important differences clearly articulated, or why this extension should be considered a challenging problem.

The paper emphasizes what it calls a new Arithmetic Mean $\\mu P$ regime, but I don't see how it is importantly different from the alternatives, and I don't even quite see how it is effectively implemented in the paper. While the paper discusses in detail the (more or less standard) layer-dependent initialization, it doesn't say anything about a specific layer-dependent learning rate schedule, which I would expect to characterize a new learning regime. It appears that this AM $\\mu P$ regime is merely used in Theorems 1-2 to justify the choice of optimal learning rate by averaging the variance of updates across the layers. However, this variance in the proofs is given by a particular cubic function of the layer index $l$ (line 809). Even if we don't average, but just consider the variance at the last layer $l=L$, or in the middle of the network, $l=L/2$, this will still produce the desired $L^{-3/2}$ scaling of the optimal learning rate. So, the role of the AM $\\mu P$ regime in the established results is not clear to me.

Moreover, the proposed AM $\\mu P$ regime includes non-uniform scenarios that look unfavorable. For example, with respect to the proposed $\overline S$ characteristic, training only the output layer of the network with a huge update variance appears equivalent to uniformly training all layers. However, the former option is likely suboptimal and conflicts with the general $\\mu P$ logic of equidistribution of learning over the layers.

**Issues with exposition / clarity**

Theorems 1-3 don't look like theorems in the proper mathematical sense. They are stated very informally, and the paper doesn't provide fully rigorous versions (even in the appendix). For example, in Theorem 1 the precise meaning of the main object, the "*learning rate scale that preserves width-invariant training dynamics*" is unclear. It appears from the proof that the theorem concerns the first step of the gradient descent after network initialization by independent weights; also, this step seems to be considered in a linear approximation. The precise setting can only be guessed by studying the proofs. "*$\\kappa$ depends only on the activation/initialization fixed point*" - unclear; no fixed point has been mentioned elsewhere. Theorem 1 seems to contradict Theorem 2: e.g., the former seems to claim that $\\eta^*$ does not depend on the channel widths $C_l$, but the latter seems to admit such dependence.

The sctructure of the paper is generally not optimal. Some points seem to be excessively repeated, while other essential information is missing. In particular, variants of the formula $\\eta^*(L)\propto L^{-3/2}$ are repeated numerous times (lines 66, 73, 96, 141, 311, 349, 360, 472), but the paper never explains in simple terms the origin of the exponent $-3/2$. The difference between $\\propto$ and $\\Theta$ in lines 66 and 73 is not explained.

**Questions:**

N/A

---

### Meta-Review · Area_Chair_4aQC · 2025-12-20

**Summary:**

This paper proposes a new parameterization regime, Arithmetic-Mean $\mu$P (AM $\mu$P), to derive a unified depth-learning-rate scaling law ($\eta^* \propto L^{-3/2}$) for Convolutional Neural Networks (CNNs) and Residual Networks (ResNets). The authors aim to extend the $\mu$P framework, which was primarily developed for homogeneous Multi-Layer Perceptrons (MLPs), to more heterogeneous modern architectures.

The paper received three reviews with a highly negative consensus: **[4, 2, 2]**. Given the low scores and the absence of an author rebuttal in the discussion forum, the decision is a clear rejection.

The paper suffers from fundamental issues across three categories: **limited novelty**, **lack of technical rigor/clarity**, and **questionable technical validity** for ResNets. The authors failed to adequately position their work against existing literature, and the presentation issues are severe enough to prevent a proper technical assessment.

**Reviewer Concerns:**

Since no author rebuttal was provided, **all concerns are considered outstanding**. The decision to reject is based on the cumulative weight of these unaddressed issues.

**Reviewer Scores:**

Since no rebuttal was submitted, no reviewer had the opportunity to participate in a discussion phase.

---

### Decision · Program_Chairs · 2026-01-26

Reject